image processing/fluid mechanics/computational mathematics

image inpainting, Cahn–Hilliard equation, finite-volume scheme, MINST dataset, damaged-image prediction

**Author for correspondence:**
Sergio P. Perez
e-mail: sergio.perez15@imperial.ac.uk

# Enhancement of damaged-image prediction through Cahn–Hilliard image inpainting

José A. Carrillo[1], Serafim Kalliadasis[2], Fuyue Liang[2] and Sergio P. Perez[2,3]

[1]Mathematical Institute, University of Oxford, Oxford OX2 6GG, UK
[2]Department of Chemical Engineering, and [3]Department of Mathematics, Imperial College London, London SW7 2AZ, UK

SPP, 0000-0001-8485-609X

We assess the benefit of including an image inpainting filter before passing damaged images into a classification neural network. We employ an appropriately modified Cahn–Hilliard equation as an image inpainting filter which is solved numerically with a finite-volume scheme exhibiting reduced computational cost and the properties of energy stability and boundedness. The benchmark dataset employed is Modified National Institute of Standards and Technology (MNIST) dataset, which consists of binary images of handwritten digits and is a standard dataset to validate image-processing methodologies. We train a neural network based on dense layers with MNIST, and subsequently we contaminate the test set with damages of different types and intensities. We then compare the prediction accuracy of the neural network with and without applying the Cahn–Hilliard filter to the damaged images test. Our results quantify the significant improvement of damaged-image prediction by applying the Cahn–Hilliard filter, which for specific damages can increase up to 50% and is advantageous for low to moderate damage.

## 1. Introduction

Image inpainting consists in filling damaged or missing areas of an image, with the ultimate objective of restoring it and making it appear as the true and original image. There are multiple applications of image inpainting, ranging from restoration of the missing areas of oil paintings and removal scratches in photographs to noisy MRI scans and blurred satellite images of the earth. Manual image inpainting techniques have been employed for many centuries by art conservators and

professional restorers, but it was not until the turn of the twenty-first century that digital image inpainting models based on partial-differential equations (PDEs) and variational methods were introduced [1–3]. These methods are usually referred to as non-texture, geometrical or structural inpainting since they focus on restoring the structural information in the inpainted domain such as edges, corners or curvatures. This is done by performing an image interpolation of the damaged areas based on the information collected from the surrounding environment only, leading to appealing images for the human vision system. On the contrary, texture inpainting is based on recovering global patterns of the image for the inpainted region [4], and a popular tool in this category is the exemplar-based inpainting methods [5,6]. Associated with these developments, a field that has gained a lot of traction in recent years is the so-called generative image inpainting, where deep-learning-based approaches have proven to be successful even for blind inpainting in which the inpainted region is not provided *a priori* [7–9]. In this work, we focus on non-texture image inpainting methods based on partial differential equations (PDEs), and we refer the reader to [10] for a general review of the topic.

There have been multiple PDE models for image inpainting proposed since the initial work of Bertalmio *et al.* [1] nearly 20 years ago. Their trail-blazing model is able to propagate isotopes, i.e. contours of uniform greyscale image intensity, through the inpainted region, a common technique employed by museum artists in restoration. It also turns out that the original model bears close connection to fluid dynamics through the Navier–Stokes equation with the image intensity function acting as the stream function [11]. Another fluid dynamic model that has played a pivotal role in image inpainting is the Cahn–Hilliard (CH) equation, initially proposed in [12] for phase separation in binary alloys. It has been employed in a wide spectrum of applications from wetting phenomena [13,14] and polymer science [15] to tumour growth [16]. This equation satisfies a gradient-flow structure employing a $H^{-1}$ norm,

$$\frac{\partial \phi(x, t)}{\partial t} = \nabla \cdot \left( M(\phi) \nabla \frac{\delta \mathcal{F}[\phi]}{\delta \phi} \right), \tag{1.1}$$

where $\phi$ is an order parameter widely referred to as the phase field which for a binary system takes on the value $\phi = 1$ in one of the phases and $\phi = -1$ in the other, while varying smoothly in the interface region with a width of $O(\epsilon)$, with $\epsilon$ being a positive parameter related to the interface thickness coming from the derivation of diffuse-interface models [17]. $M(\phi)$ is the mobility obtained here from the one-sided model $M(\phi) = 1$ and $\mathcal{F}[\phi]$ is the free energy satisfying

$$\mathcal{F}[\phi] = \int_{\Omega} \left( H(\phi) + \frac{\epsilon^2}{2} |\nabla \phi|^2 \right) dx, \tag{1.2}$$

with the variation of the free energy denoted as $\xi$ and given by

$$\xi = \frac{\delta \mathcal{F}[\phi]}{\delta \phi} = H'(\phi) - \epsilon^2 \Delta \phi, \tag{1.3}$$

and $H(\phi)$ taken here as the Ginzburg–Landau double-well potential with the two wells corresponding to the two phases,

$$H(\phi) = \frac{1}{4}(\phi^2 - 1)^2 \quad \text{for } \phi \in [-1, 1]. \tag{1.4}$$

The boundary conditions imposed for the CH equation in (1.1) are no-flux for both the phase field and for the variation of the free energy,

$$\epsilon^2 \nabla \phi \cdot \boldsymbol{n} = 0, \quad M(\phi) \nabla \xi \cdot \boldsymbol{n} = 0, \tag{1.5}$$

where $\boldsymbol{n}$ is defined in the normal direction to and into the wall.

There is a clear physical interpretation of the terms forming the free energy in (1.2): on the one hand, the potential $H(\phi)$ contains the equilibrium information of the system, and each of the minima of the choice (1.4) correspond to an equilibrium phase; on the other hand, the gradient term in (1.2) accounts for the cost of spatial inhomogeneities, resulting in a surface tension between different phases. As a result, the equilibrium state balances the tendency of separation from the hydrophobic potential $H(\phi)$ and the tendency of mixing from the hydrophilic gradient term. The mobility term impacts the rate of phase separation and the coarsening process, and some works employ non-constant degenerate mobilities that cancel when the phase field corresponds to one of the two wells in (1.4) (see e.g. [18–20] for further discussions of the physical significance of the various terms of the CH equation).

The CH equation was firstly proposed in the context of image inpainting by Bertozzi *et al.* [21]. Specifically, the authors adopted a modified CH equation for binary images with inpainting quality as accurate as the state-of-the-art inpainting models but with a much faster computational speed taking advantage of the efficient computational techniques already available for the CH equation (DJ Eyre 1998, unpublished article) [22]. Since then several authors have extended the applicability of the CH equation in the field of image inpainting, for instance by taking into account grey-value images [23,24], non-smooth potentials instead of the double-well potential (1.4) [25], and considering colour image inpainting [26]. The modified CH equation in [21] introduces a fidelity term $\lambda(x)$ to avoid modifying the original image outside of the inpainted region $D$, and the CH equation in (1.1) becomes

$$\frac{\partial \phi(x, t)}{\partial t} = -\nabla^2(\epsilon^2 \nabla^2 \phi - H'(\phi)) + \lambda(x)(\phi(x, t=0) - \phi), \tag{1.6}$$

where

$$\lambda(x) = \begin{cases} 0 & \text{if } x \in D, \\ \lambda_0 & \text{if } x \notin D, \end{cases} \tag{1.7}$$

and $\phi(x, t=0)$ refers to the original damaged image. The parameter $\epsilon$ plays a similar role as in the original CH equation, and here it is related to the interface between the two phases or colours presented in the image. The two parameters $\epsilon$ and $\lambda_0$ are essential to achieve an adequate image inpainting outcome, and it is usually necessary to iterate until finding appropriate tunings for their values, which typically depend on the image specifications. Furthermore, the new modified CH equation (1.6) is not strictly a gradient flow: although the original CH equation satisfies a gradient-flow structure under an $H^{-1}$ norm and the fidelity term in (1.6) can be derived from a gradient flow under an $L^2$ norm, the combined modified CH equation is neither a gradient flow in $H^{-1}$ nor $L^2$.

One of the main advantages of employing the CH equation for image inpainting is the myriad of fast and reliable numerical methods available for its solution. A pivotal contribution was the convex-splitting scheme developed by DJ Eyre (1998, unpublished article), which is unconditionally energy-stable, by treating as implicit the convex terms of the free energy in (1.2), while keeping the concave terms explicit. The design of energy-stable and maximum-principle satisfying schemes for the CH equation has been an active area of research [27], and several authors have proposed schemes based on finite differences [28–30], finite elements [31,32], finite volumes [33] or discontinuous Galerkin [34–37]. These schemes have also proven effective for degenerate mobilities or logarithmic potentials. Schemes satisfying the maximum principle condition for specific choices of free energy have also been constructed [38]. We refer the reader to [39] for a recent work discussing the state-of-the-art numerical techniques for nonlinear gradient flows.

In a recent effort [19], we constructed a robust semi-implicit finite-volume scheme for the CH equation that offers crucial advantages when applied to the field of image inpainting:

— Firstly, finite volumes are a straightforward discretization when dealing with images, which often consist of rectangular-shaped pixel cells with an average colour intensity. This is exactly the starting point of finite-volume schemes, and as a result it is conceptually simpler to apply finite volumes in comparison with finite elements, finite differences or discontinuous Galerkin (which would be more suitable for other more complex and rare pixel shapes such as triangular ones).

— Secondly, our scheme is based on a dimensional-splitting approach: instead of solving the full two-dimensional image altogether, dimensional splitting initially solves row by row and then column by column. This has a massive benefit in computational cost, which is reduced from $\mathcal{O}(N^{d\gamma})$ for an image with $N$ cells in $d$ dimensions to $\mathcal{O}(dN^{d+\gamma-1})$, with $2 < \gamma < 3$. The reason for this is that the cost of inverting an $N \times N$ matrix is $\mathcal{O}(N^{d\gamma})$, with $2 < \gamma < 3$ that slightly varies depending on the inversion algorithm and matrix structure (see [40] for details). For images with $N$ cells per dimension, the solution of the full two-dimensional scheme involves inverting a $N^d \times N^d$ Jacobian matrix, with a subsequent cost of $\mathcal{O}(N^{d\gamma})$. By contrast, the dimensional-splitting technique requires inverting $dN^{d-1}$ Jacobians of size $N \times N$, amounting for a total computational cost of $\mathcal{O}(dN^{d+\gamma-1})$. This is already advantageous for a two-dimensional image, and the computational cost is further reduced for high-dimensional images, such as the ones for [41] or X-ray computed tomography [42] in medical image analysis. To add more, the dimensional-splitting technique allows for parallelization, and it is possible to halve the total computational cost by solving non-adjacent rows and columns in parallel.

— Thirdly, our scheme has been extensively tested in [19] for challenging configurations of the original CH equation (1.1). In addition, in [19] we prove that the scheme unconditionally satisfies the discrete

decay of the free energy for different choices of potentials (1.4) [25], while at the same time we prove the phase-field boundedness for mobilities of the form $M(\phi) = 1 - \phi^2$. Even though the modified CH equation (1.6) is not strictly a gradient flow due to the inclusion of the fidelity term, our scheme preserves its robustness for all the image-inpainting test cases presented in this work.

The combination of these properties and reduced computational cost, together with the versatility of finite volumes, make our scheme efficient and robust for the solution of the modified CH equation in (1.6) for a variety of applications in image inpainting.

The objective of this work is to show precisely the applicability of our numerical framework in [19] for a benchmark dataset of images in need of restoration through image inpainting. For this task, we purposely add different types and intensities of damage to the popular Modified National Institute of Standards and Technology (MNIST) dataset [43], and then apply image inpainting by solving the modified CH equation (1.6) with our finite-volume scheme in [19]. The MNIST database is a standard dataset to validate image-processing methodologies, and it contains binary handwritten images of numbers from zero to nine. We choose this dataset since the modified CH equation (1.6) is applicable to binary images such as the ones in the MNIST dataset. The extension of this work to non-binary images is also relevant and will be explored elsewhere. We also assess the improvement in pattern recognition accuracy of the restored MNIST images, and for this we construct a neural network for the task of classification. A key objective of our study is to quantify the benefits of including a CH filter before introducing a damaged image into a neural network. Our results demonstrate that accuracies in classification can increase up to 50% for particular damages in the images, and, in general, applying the CH filter improves the accuracy prediction for a wide range of low to moderate image damage. This increase in accuracy is obtained by applying our image-inpainting methodology to the MNIST dataset exclusively, and as a disclaimer we remark that other datasets or types of damage may result in different increases of accuracy. The application of other types of filters, such as texture inpainting or generative inpainting, may also result in different increases of accuracy, but overall the accuracy should be higher compared to the case where no filter is applied to the damaged image. The code to reproduce the results of this work is available at [44].

In §2, we outline the methodology: in §2.1 we adapt our finite-volume scheme in [19] for the modified CH equation in (1.6); in §2.2, we recall the two-step method for image inpainting in [21]; in §2.3, we detail the neural network architecture for the classification task; and lastly in §2.4, we explain the structure of the integrated algorithm which takes a damaged image, applies a CH filter to it, and then classifies the image through a neural network. Subsequently in §3, we present the results of the integrated algorithm applied to the MNIST dataset: in §3.1, we begin by identifying appropriate tunings for the values of $\epsilon$ and $\lambda_0$; in §3.2, we present the different types of damage introduced into the MNIST testset of images; and finally in §3.3, we quantify the improvement in accuracy of applying the CH filter to the damaged MNIST images before introducing them into the neural network. A discussion and final remarks are offered in §4.

# 2. Integrated algorithm with image inpainting and pattern recognition

We detail the construction of an integrated algorithm that firstly applies image inpainting and subsequently conducts pattern recognition for the restored image. In §2.1, we begin by presenting the finite-volume scheme employed to solve the modified CH equation, based on the work in [19]. Then in §2.2 we illustrate the two-step method for image inpainting, based on tuning the parameters $\epsilon$ and $\lambda_0$ of the modified CH equation. In §2.3, we present the neural network employed for pattern recognition, detailing its architecture and training parameters. Finally, in §2.4, we gather all previous elements to formulate an integrated algorithm for prediction with an image-inpainting filter.

## 2.1. Two-dimensional finite-volume scheme for the modified CH equation

We summarize the two-dimensional finite-volume scheme constructed for the original CH equation in our previous work [19]. This scheme satisfies an unconditional decay of the discrete free energy of the original CH equation in (1.2), independently of the time step, and for specific choice of mobility $M(\phi) = 1 - \phi^2$ it ensures the unconditional boundedness of the phase field. The scheme can be straightforwardly extended to the modified CH equation in (1.6) proposed in [21] as we show here. Even though the modified CH equation does not possess some of the properties of the original CH

equation, such as the gradient-flow structure, our finite-volume scheme preserves its robustness for all the image-inpainting test cases presented in this work. In remark 2.1, we explain how to choose the time step and the mesh size, and in remark 2.2, we detail how to turn the scheme into a dimensional-splitting one, with promising applicability in high-dimensional images such as medical ones. We refer the reader to [19,45] for further details about dimensional-splitting schemes.

For simplicity, let us rewrite (1.6) in two dimensions by introducing $u = (v, w)$ as the physical velocity term and $\xi$ as the variation of the free energy with respect to the density,

$$\frac{\partial \phi(x, y, t)}{\partial t} = -\nabla \cdot u + \lambda(x, y)(\phi(x, y, t = 0) - \phi(x, y, t)), \tag{2.1}$$

where

$$\xi = \frac{\delta \mathcal{F}[\phi]}{\delta \phi} = \epsilon^2 \nabla^2 \phi - H'(\phi), \quad u = \nabla \xi, \quad v = \frac{\partial \xi}{\partial x}, \quad \text{and} \quad w = \frac{\partial \xi}{\partial y}. \tag{2.2}$$

For the finite-volume formulation, we begin by dividing the computational domain $[0, L] \times [0, L]$ in $N \times N$ cells $C_{i,j} := [x_{i-1/2}, x_{i+1/2}] \times [y_{j-1/2}, y_{j+1/2}]$, all with uniform size $\Delta x \Delta y$ so that $x_{i+1/2} - x_{i-1/2} = \Delta x$ and $y_{j+1/2} - y_{j-1/2} = \Delta y$. The time step is denoted as $\Delta t$. In each of the cells we define the cell average $\phi_{i,j}^n$ at time $t = n \Delta t$ as

$$\phi_{i,j}^n = \frac{1}{\Delta x \Delta y} \int_{C_{i,j}} \phi(x, y, t = n\Delta t) \, dx dy, \tag{2.3}$$

where $\phi_{i,j}^0$ is the phase field at $t = 0$, which corresponds to the normalized pixel intensities of the initial damaged image to be inpainted.

The finite-volume scheme is then derived by integrating the modified CH equation (2.1) over each of the cells $C_{i,j}$ of the domain, leading to an implicit formulation satisfying

$$\frac{\phi_{i,j}^{n+1} - \phi_{i,j}^n}{\Delta t} = -\frac{F_{i+(1/2),j}^{n+1} - F_{i-(1/2),j}^{n+1}}{\Delta x} - \frac{G_{i,j+(1/2)}^{n+1} - G_{i,j-(1/2)}^{n+1}}{\Delta y} + \lambda_{i,j}(\phi_{i,j}^0 - \phi_{i,j}^n), \tag{2.4}$$

with $F_{i+1/2,j}^{n+1}$ and $G_{i,j+1/2}^{n+1}$ being flux approximations at the boundaries, and $\lambda_{i,j}$ the discrete version of $\lambda(x)$ in (1.7) satisfying

$$\lambda_{i,j} = \begin{cases} 0 & \text{if } (x_i, y_j) \in D, \\ \lambda_0 & \text{if } (x_i, y_j) \notin D, \end{cases} \tag{2.5}$$

with $D$ being the inpainted domain where the image damage is located. The impainted domain has to be determined beforehand for each image, and is formed by those finite-volume cells to be repaired during the image inpainting.

The flux terms $F_{i+1/2,j}^{n+1}$ and $G_{i,j+1/2}^{n+1}$ are the discrete approximations of the velocity components $v$ and $w$ at the cell interfaces $(x_{i+1/2}, y_j)$ and $(x_i, y_{j+1/2})$, respectively. Such approximations follow an upwind and implicit approach inspired by the works in [19,45–47], satisfying

$$\left. \begin{aligned} F_{i+(1/2),j}^{n+1} &= \left(v_{i+1/2,j}^{n+1}\right)^+ + \left(v_{i+1/2,j}^{n+1}\right)^- \\ G_{i,j+\frac{1}{2}}^{n+1} &= \left(w_{i,j+1/2}^{n+1}\right)^+ + \left(w_{i,j+1/2}^{n+1}\right)^-, \end{aligned} \right\} \tag{2.6}$$

and

where the velocities $v_{i+(1/2),j}^{n+1}$ and $w_{i,j+\frac{1}{2}}^{n+1}$ are discretized from (2.2) as

$$v_{i+(1/2),j}^{n+1} = -\frac{\xi_{i+1,j}^{n+1} - \xi_{i,j}^{n+1}}{\Delta x} \quad \text{and} \quad w_{i,j+\frac{1}{2}}^{n+1} = -\frac{\xi_{i,j+1}^{n+1} - \xi_{i,j}^{n+1}}{\Delta y}, \tag{2.7}$$

where $\xi_{i,j}^{n+1}$ is the discretized variation of the free energy defined in (2.2). The upwind approach in (2.6) follows from

$$\left. \begin{aligned} \left(v_{i+1/2,j}^{n+1}\right)^+ &= \max(v_{i+1/2,j}^{n+1}, 0), \quad \left(v_{i+1/2,j}^{n+1}\right)^- = \min(v_{i+1/2,j}^{n+1}, 0), \\ \left(w_{i,j+1/2}^{n+1}\right)^+ &= \max(w_{i,j+1/2}^{n+1}, 0), \quad \left(w_{i,j+1/2}^{n+1}\right)^- = \min(w_{i,j+1/2}^{n+1}, 0). \end{aligned} \right\} \tag{2.8}$$

and

The discretized variation of the free energy $\xi_{i,j}^{n+1}$ in (2.2) is computed with a semi-implicit scheme inspired by the ideas of DJ Eyre (1998, unpublished article) [22], where the so-called convexity splitting scheme is proposed to construct unconditional gradient-stable schemes (i.e. schemes that

ensure the decay of the discrete version of the free energy in (1.2)). In our recent effort [19], we show that our finite-volume scheme unconditionally decreases the discrete free energy of the CH equation if the contractive part of the potential, $H_c(\phi)$, is taken as implicit; the expansive part of the potential, $H_e(\phi)$, is taken as explicit; and the Laplacian is taken as an average between the explicit and the implicit second-order discretizations, so that

$$
\left.
\begin{aligned}
& \xi_{i,j}^{n+1} = H_c'(\phi_{i,j}^{n+1}) - H_e'(\phi_{i,j}^{n}) - \frac{\epsilon^2}{2}\left[(\Delta\phi)_{i,j}^{n} + (\Delta\phi)_{i,j}^{n+1}\right] \\
\text{and} \qquad & H(\phi) = H_c(\phi) - H_e(\phi) = \frac{\phi^4+1}{4} - \frac{\phi^2}{2},
\end{aligned}
\right\}
\tag{2.9}
$$

where the potential $H(\phi)$ in (1.4) is decomposed into two convex functions, $H_c(\phi)$ and $H_e(\phi)$,

$$
H(\phi) = H_c(\phi) - H_e(\phi) = \frac{\phi^4+1}{4} - \frac{\phi^2}{2}.
$$

The discrete two-dimensional approximation of the Laplacian $(\Delta\phi)_{i,j}^{n}$ is chosen to satisfy the second-order form

$$
(\Delta\phi)_{i,j}^{n} := \frac{\phi_{i+1,j}^{n} = -2\phi_{i,j}^{n} + \phi_{i-1,j}^{n}}{\Delta x^2} + \frac{\phi_{i,j+1}^{n} - 2\phi_{i,j}^{n} + \phi_{i,j-1}^{n}}{\Delta y^2}.
\tag{2.10}
$$

The modified CH equation in (1.6) employs the no-flux boundary conditions defined in (1.5). The numerical implementation of the boundary conditions follows from

$$
\left.
\begin{aligned}
& \phi_{i-1,j}^{n} = \phi_{i,j}^{n} \quad \text{for } i=1,\ \forall j; \qquad \phi_{i+1,j}^{n} = \phi_{i,j}^{n} \quad \text{for } i=N,\ \forall j; \\
& \phi_{i,j-1}^{n} = \phi_{i,j}^{n} \quad \text{for } j=1,\ \forall i; \qquad \phi_{i,j+1}^{n} = \phi_{i,j}^{n} \quad \text{for } j=N, \forall i; \\
& F_{i-(1/2),j}^{n+1} = 0 \quad \text{for } i=1,\ \forall j; \qquad F_{i+(1/2),j}^{n+1} = 0 \quad \text{for } i=N,\ \forall j \\
\text{and} \qquad & G_{i,j-\frac{1}{2}}^{n+1} = 0 \quad \text{for } j=1, \forall i; \qquad G_{i,j+\frac{1}{2}}^{n+1} = 0 \quad \text{for } j=N,\ \forall i.
\end{aligned}
\right\}
\tag{2.11}
$$

**Remark 2.1 (Choice of time step $\Delta t$ and mesh size $\Delta x = \Delta y$).** Explicit finite-volume schemes are often stable under a Courant–Friedrichs–Lewy (CFL) condition imposed over the time step and mesh size. The implicit scheme presented in §2.1 does not need a CFL condition to decay the discrete free energy of the original Cahn–Hilliard equation in (1.1), as proved in [19]. However, this does not mean that any $\Delta t$ works in practice: in each time step, the nonlinear system (2.4) has to be solved by iteration, and with a large $\Delta t$ the convergence is likely to fail. In this work we have tested that with the choice of $\Delta t = 0.1$ and $\Delta x = \Delta y = 1$ our simulations converge to the right solution for the simulations presented in §3.

**Remark 2.2 (Dimensional-splitting scheme and parallelization).** The full two-dimensional finite-volume scheme presented in §2.1 can be reformulated by employing a dimensional-splitting methodology: instead of solving the full two-dimensional image altogether, this technique initially solves row by row and then column by column. The detailed construction of such a scheme is presented in [19] and is based on [45]; and here we briefly illustrate how it works.

For the dimensional-splitting approach, we firstly update the solution along the $x$-axis, for each index $j$ corresponding to a fixed value of $y_j$ where $j \in [1, N]$. Subsequently, we proceed in the same way along the $y$-axis, for each index $i$ corresponding to a fixed value of $x_i$, where $i \in [1, N]$. The index $r$, where $r \in [1, N]$, denotes the index $j$ of the fixed $y_j$ value in every $x$-axis of the first loop, and the updated average density for each $x$-axis with $j = r$ is $\phi_{i,j}^{n,r}$. Similarly, the index $c \in [1, N]$ denotes the index $i$ for every fixed value of $x_j$ in each $y$-axis of the second loop, and the updated density for each $y$-axis with $i = c$ is $\phi_{i,j}^{n,c}$.

In the first place, we go through each of the $x$-axes of the domain at a fixed $y_j$ with $j = r$. The initial conditions for the scheme are $\phi_{i,j}^{n,0} := \phi_{i,j}^{n}$. The scheme for each $x$ along the loop $r = 1, \ldots, N$ satisfies

$$
\phi_{i,j}^{n,r} =
\begin{cases}
\phi_{i,j}^{n,r-1} - \frac{\Delta t}{\Delta x}\left(F_{i+1/2,j}^{n,r} - F_{i-(1/2),j}^{n,r}\right) + \lambda_{i,j}(\phi_{i,j}^{0} - \phi_{i,j}^{n,r-1}) & \text{if } j = r; \\
\phi_{i,j}^{n,r-1} & \text{otherwise;}
\end{cases}
\tag{2.12}
$$

with $F_{i+1/2,j}^{n,r}$ computed in a similar fashion as in (2.6). Once the loop for the rows is completed, we define the intermediate density values as $\rho^{n+1/2} := \rho^{n,N}$. Subsequently, we continue through each of the $y$-axes with index $c = 1, \ldots, N$, each of them at a fixed $x_i$ with $i = c$. The initial condition for this scheme is

$\phi_{i,j}^{n,0} := \phi_{i,j}^{n+1/2}$. The scheme for each $y$ along the loop $c = 1, \ldots, N$ satisfies:

$$\phi_{i,j}^{n,c} = \begin{cases} \phi_{i,j}^{n,c-1} - \frac{\Delta t}{\Delta y}\left(G_{i,j+1/2}^{n,c} - G_{i,j-\frac{1}{2}}^{n,c}\right) + \lambda_{i,j}(\phi_{i,j}^0 - \phi_{i,j}^{n,c-1}) & \text{if } i = c; \\ \phi_{i,j}^{n,c-1} & \text{otherwise.} \end{cases} \quad (2.13)$$

Once the loop for the columns is completed, we define the final density values $\phi_{i,j}^{n+1}$ after a discrete timestep $\Delta t$ as $\phi_{i,j}^{n+1} := \phi_{i,j}^{n,N}$. This dimensional-splitting scheme can be fully parallelized in order to save computational time. This is because: (i) the scheme does not take notice of the order of updating the rows/columns, as long as all of them are updated; (ii) one row or column only depends on the values of the directly adjacent rows or columns, respectively. As a result, a strategy to parallelize the dimensional-splitting scheme consists of updating at the same time all the odd rows/columns, since they do not depend on one another. One can then proceed with all the even rows/columns at the same time.

## 2.2. Two-step method for the modified CH equation

The two-step method followed here was firstly proposed in [21]. It basically consists of applying the finite-volume scheme in §2.1 twice with different values of the parameter $\epsilon$. The first stage consists of taking a large $\epsilon$ to execute a large-scale topological reconnection of shapes, leading to images with diffused edges. Subsequently, and in order to sharpen the edges after the first stage, $\epsilon$ is substantially reduced, and the final outcome becomes less blurry and diffused. We denote the corresponding values of $\epsilon$ as $\epsilon_1$ and $\epsilon_2$.

Adequately tuning the two values of $\epsilon$ in each stage, as well as $\lambda$, is vital to complete a successful image inpainting. Those values have to be chosen empirically and depend on the dataset and type of damage, and in §3.1 we conduct a study to select them (with a sensitivity analysis provided in appendix A). As explained there, the appropriate values for $\epsilon$ are between 0.5 and 1.5 for MNIST-like images, while $\lambda \in [1, 1000]$. The cell sizes are $\Delta x = \Delta y = 1$. The reader can find the exact values employed after the analysis of §3.1 in table 2.

## 2.3. Neural network architecture for classification

The prediction of the label in the restored images is performed via a neural network constructed in TensorFlow [48]. Its architecture is defined taking into account that in this work we employ the MNIST dataset [43], which contains binary images of digits from 0 to 9 and has a resolution of $28 \times 28$ pixels. This is a benchmark dataset in the community and is the de facto 'hello world' dataset of computer vision. There are consequently plenty of neural network architectures attaining extremely high accuracies for the MNIST dataset, and we refer the reader to the Kaggle competition of Digit Recognizer in [49] for examples of such architectures.

Here, however, our overarching objective is to quantify how the prediction of damaged images is enhanced once the CH filter is applied to the images beforehand. Hence, we do not require a highly sophisticated neural network and a cutting-edge architecture as in computer vision: our images are not going to be exactly the same as in the training set due to the damage and the subsequent restoration. We then select a standard architecture for classification based on sequential dense layers. Such architecture is formed by:

(1) A flatten layer that takes the $28 \times 28$ image input and turns it into an array with 784 elements. There are no weights to optimize in this layer.
(2) A dense layer with 64 units and the rectified linear unit (ReLU) activation function, defined as $f(x) = \max\{0, x\}$ for $x \in \mathbb{R}$. There are $784 \times 64$ weights to optimize in this layer, in addition to the bias term in each of the 64 units.
(3) Another dense layer with 64 units and the ReLU activation function. There are $64 \times 64$ weights to optimize in this layer, in addition to the bias term in each of the 64 units.
(4) A final dense layer with 10 units and the softmax activation function, which returns the normalized probability distribution for the 10 labels and satisfies $\sigma(z_i) = \exp(z_i) / \sum_{j=1}^{10} \exp(z_j)$, with $z = (z_1, \ldots, z_{10})$ being the output of the final dense layer with 10 units. There are $64 \times 10$ weights to optimize in this layer, in addition to the bias term in each of the 10 units.

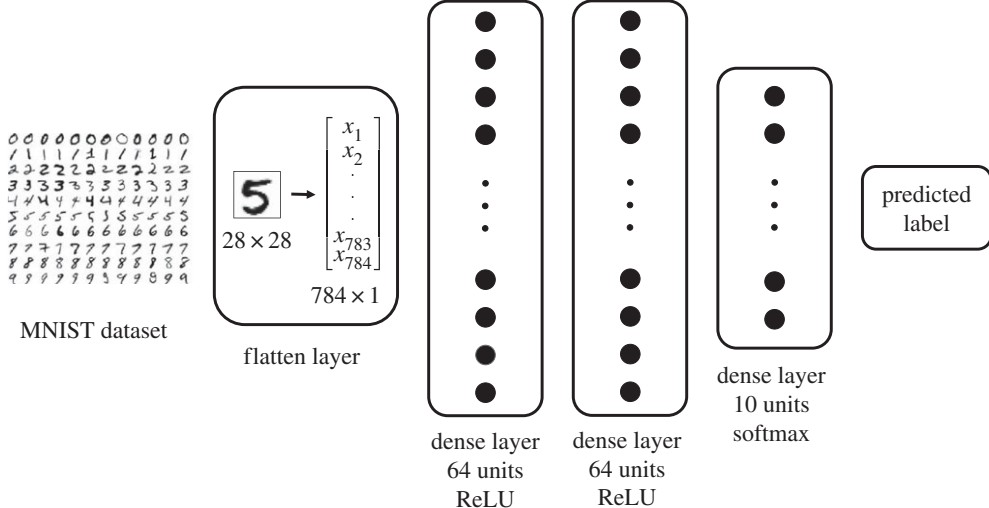

**Figure 1.** Diagram showing the layers of the neural network of §2.3.

For the training of this network, we initially divide the original MNIST dataset in 60 000 training images and 10 000 testing images. We then train the neural network for 10 epochs with the Adam optimizer, choosing as loss function the categorical cross-entropy defined as

$$J(w) = \frac{1}{N} \sum_{i=1}^{N} [y_i \log(\hat{y}_i) + (1 - y_i) \log(1 - \hat{y}_i)],$$

with $w$ being the weights to optimize, $y_i$ each of the $N$ true labels of the training dataset, and $\hat{y}_i$ each of the $N$ predicted labels. After 10 epochs, we get an accuracy for the training dataset of 99.02%, while for the test set the accuracy is 97.47%. Once the neural network is trained, we keep the weights fixed for the comparison of damaged and restored images in §3. A display of the neural network is depicted in figure 1.

## 2.4. Integrated algorithm

The integrated algorithm proposed and tested in this work takes as input a damaged image, applies the CH filter based on §2.1 and §2.2 to restore it, and finally applies the already-trained neural network in §2.3 to predict its label.

   To show the applicability of this integrated algorithm, we initially create damage in the images of the test set in the MNIST dataset [43]. After we apply the image inpainting to the damaged test images, we introduce the restored images in the neural network. At that point, and since we have the true labels of the test set, we can assess the attained accuracy in comparison to directly introducing the damaged images or the original images into the neural network. This procedure is conducted for multiple types of damage in §3, and a schematic of all steps is depicted in figure 2.

# 3. Application of the integrated algorithm to the MNIST dataset

Our focus here is testing the applicability of the integrated algorithm in §2.4 to increase predictability in damaged images. In §3.1, we start by analysing the impact of the parameters $\lambda_0$ and $\epsilon$ on the inpainting process, with the objective of calibrating them before employing the MNIST dataset. In §3.2, we detail the types of damage that we insert into the MNIST dataset, and we also show the restored outcomes of applying the CH equation as an image inpainting filter. Finally, in §3.3, we evaluate how the accuracy of the damaged images increases after applying the CH filter to them, for various types and degrees of damage.

   The code to reproduce the results of this work is available at [44].

## 3.1. Inpainting of a cross-line

We employ the cross-line example in [21] to analyse the role of the parameters $\epsilon$ and $\lambda_0$ in the finite-volume scheme of §2.1. These two parameters crucially determine the success of the image inpainting

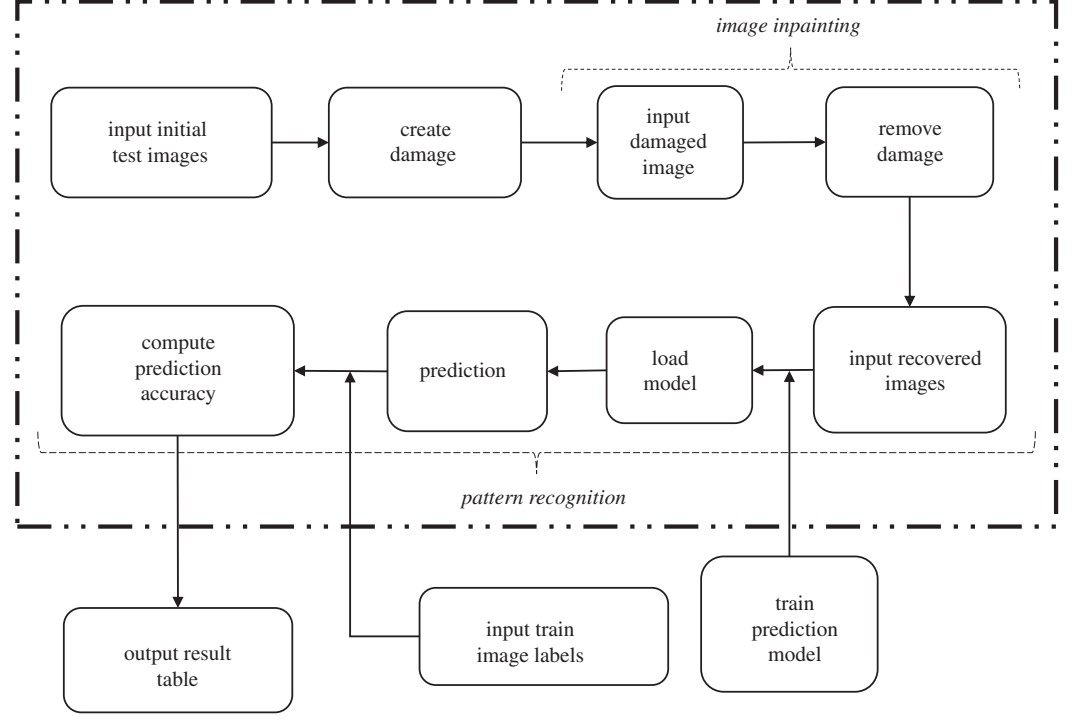

**Figure 2.** A schematic to show the applicability of the integrated algorithm.

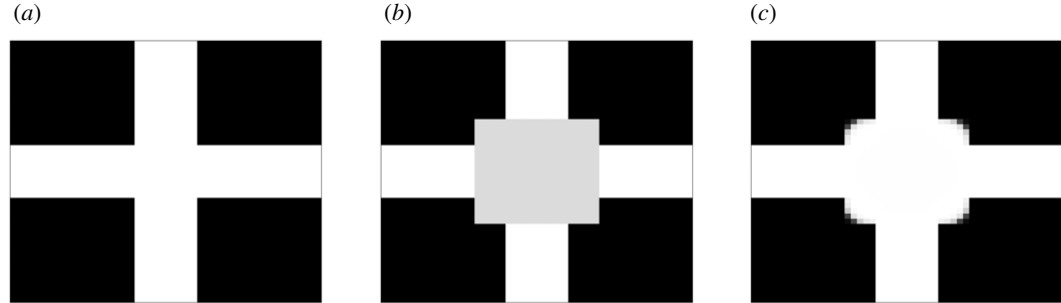

**Figure 3.** Image inpainting of a cross-line, inspired by Bertozzi *et al*. [21]. (*a*) Initial cross-line, (*b*) damaged cross-line, (*c*) result from image inpainting.

procedure, and consequently appropriate calibrations for the parameters must be chosen before running the scheme. The original cross-line image is depicted in figure 3*a*, and we add to it a grey damage in the centre, as shown in figure 3*b*. This image contains $50 \times 50$ pixels or cells, each with a size of $\Delta x = \Delta y = 1$. We apply the finite-volume scheme in §2.1 to the damaged image in figure 3*b*.

We first aim to determine $\lambda_0$ in (2.5) and we set $\epsilon = 1$ as an initial guess so that $\epsilon = \Delta x = \Delta y$. From (2.5), $\lambda_{i,j}$ is only non-zero for the predefined area of undamaged pixels. Indeed the term with $\lambda_{i,j}$ in the finite-volume scheme in (2.4) ensures that the undamaged pixels are not modified during the image inpainting, but for this $\lambda_0$ has to be sufficiently large to counterbalance the fluxes of the scheme. Bearing this in mind, we run the numerical scheme with a $\Delta t = 0.1$ and until the $L^1$ norm between successive states is lower than a certain tolerance fixed to be $10^{-4}$. Our simulation produces satisfactory results and does not break down for a range of $\lambda \in [1, 1000]$. Table 1 shows that the computational time to reach the required tolerance decreases when increasing the value of $\lambda$. In addition, the $L^1$ norm between the final and initial state is also lower for greater $\lambda$. It is worth mentioning that different choices for $\Delta t$ yield different ranges of valid $\lambda$, given that (2.4) is a singularly perturbed problem for large $\lambda_0$. Hence, greater values of $\lambda$ are possible if $\Delta t$ is refined. In our case, with the choice of $\Delta t = 0.1$, our finite-volume scheme does not yield any result and breaks down for values of $\lambda \notin [1, 1000]$.

We next consider the tuning of the parameter $\epsilon$ which in turn is related to the pixel size $\Delta x$ and $\Delta y$. For values of $\epsilon$ larger than the pixel size the outcome tends to be diffusive, while for smaller values the edges

**Table 1.** Comparison for different values of $\lambda$: computational time before reaching the tolerance and $L^1$ norm between the final and initial state.

| $\lambda$ | time | $L^1$ norm |
|---|---|---|
| 1 | 489.8 | 61.2 |
| 10 | 489.7 | 60.8 |
| 100 | 484.8 | 60.8 |
| 1000 | 481.5 | 60.7 |

**Table 2.** Optimal parameters of the two-step algorithm applied in the MNIST dataset. $\epsilon_1$ and $\epsilon_2$ denote the values of $\epsilon$ used in the first and second step, respectively, as explained in §2.2.

| parameters | | $\epsilon_1$ | $\epsilon_2$ | $\lambda$ |
|---|---|---|---|---|
| customized damage | | 1.5 | 0.5 | 1000 |
| random damage | rows | 1.5 | 0.5 | 1000 |
| | pixels | 1.5 | 0.5 | 9000 |

are sharpened. When applying the finite-volume scheme in (2.4) with $\lambda \in [1, 1000]$ we obtain satisfactory results for $\epsilon \in [0.5, 1.5]$, while for values outside this range the simulation breaks down because of the singular nature of (2.4). As a consequence, for the two-step method in §2.2 we first take the value $\epsilon_1 = 1.5$ for the large-scale topological reconnection of shapes, while for the second step we choose the value $\epsilon_2 = 0.5$ to sharpen the edges. The image impainting of the damaged image in figure 3b resulting from applying this choice of parameters is shown in figure 3c. More details about the choice of $\lambda$, $\epsilon_1$ and $\epsilon_2$ are provided in appendix A.

The final outcome after the image inpainting in figure 3b is not the same as the original image in figure 3a. The reason for that is explained in [50], where multiple steady-state solutions of the modified CH equation were shown to exist. As the information under the inpainting region has been destroyed, there is no way of knowing that the steady state we obtain is less accurate than other viable solutions, in comparison to figure 3a. For further details, we refer the reader to [50], where a bifurcation analysis is carried out to show that the steady state may vary depending on the choices for $\epsilon$ and $\Delta x$, $\Delta y$.

## 3.2. Damage introduced in the MNIST dataset

Here we discuss the types of damage inserted into the MNIST test set, with the objective of subsequently applying the CH filter developed in §2.1 for image inpainting. The varied damage employed aims to represent a mock case of damage that may be encountered in an image in need for restoration. As a result, we decide to employ two kinds of damage with different intensities: customized damage affecting particular regions of the image, and random damage selecting arbitrary pixels or horizontal lines in the image. Specifically:

(a) Customized damage: this type of damage is applied in four different fashions, as shown in figure 4. The basic idea is to turn vertical or horizontal lines of pixels into a uniform grey intensity between black and white colour. In figure 4, we show the outcome of applying the CH filter to the damaged images. It can be seen that our model is able to recover the images from the different types of damage, albeit with varying degrees of success. For instance, the damage introduced in figure 4c is a thick horizontal line implying a considerable loss of information from the original image, compared to the other types of damage. As a result, the inpainted image filter for this type of damage is not as effective as the other ones, as can be seen from the inpaintings in figure 4.

(b) Random damage: this second type of damage is inserted in a random fashion and with different levels of intensity. Two ways of randomly creating damage are considered: one makes use of randomly selecting whole horizontal rows of pixels, while the other is obtained by randomly selecting individual pixels. In addition, for both types of random damage we employ different

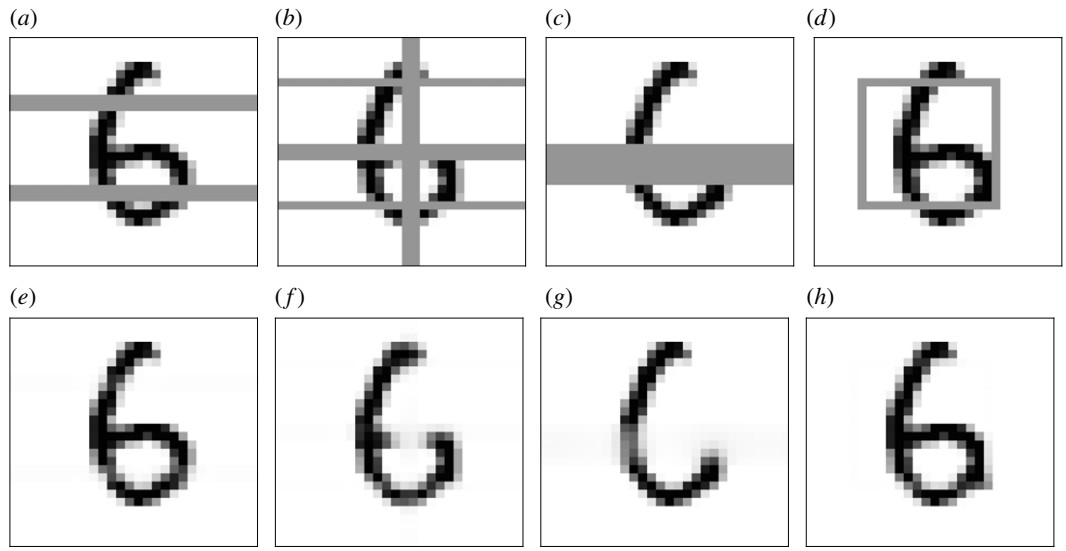

**Figure 4.** Customized damage applied to a particular sample of the MNIST dataset. ($a$–$d$): the sample with four different types of damage; ($e$–$f$): The outcome of applying image inpainting to the damaged samples.

levels of damage intensity, so that a higher percentage of the image contains damage if the intensity rises. This allows us to test how our image inpainting algorithm behaves with increasing levels of damage in the image. Examples of these damages are shown in figure 5. Similarly to the case of customized damage, the higher the intensity of damage the more information is lost in the inpainting, as we can see for example in the case of 80% pixel damage in figure 5. But despite this, our image inpainting algorithm renders recognizable images even with relative high amounts of damage.

The parameter values for $\lambda$, $\epsilon_1$ and $\epsilon_2$ are gathered in table 2, and follow the reasoning discussed in §3.2. This choice of parameters in our finite-volume scheme in §2.1 leads to an effective image-inpainting algorithm capable of restoring images with damage of varied nature as shown in figures 4 and 5. Further details about the selection of $\lambda$, $\epsilon_1$ and $\epsilon_2$ are provided in appendix A, where a sensitivity analysis is provided in order to justify the parameter choices in table 2. The next step is to integrate this image-inpainting algorithm within a pattern recognition framework for the MNIST dataset.

## 3.3. Pattern recognition for inpainted images

We now apply the neural network described in §2.3 to predict labels of damaged images with and without image inpainting, with the aim of quantifying the improvement of accuracy following the application of the CH filter. This study is completed for the different types and intensities of the noise depicted in §3.2.

We begin by adding the types of damage in §3.2 to the 10 000 samples of the MNIST test dataset. The next step is to apply the CH filter and the two-step method to each one of them, while also saving copies of the test images with the damage. Eventually, for each type of damage we get two batches of 10 000 images: one still with the damage, and another one with image inpainting applied. Given that the neural network of §2.3 is already trained with the 60 000 samples of the MNIST training dataset, we can directly compute the accuracy of each of the two batches. This way we are able to assess the improvement in accuracy thanks to applying image inpainting to restore the damage.

For the validation we just employ the accuracy metric, which is defined as follows:

$$\mathrm{accuracy} \equiv \frac{\text{number of correct predictions}}{\text{number of total predictions}}.$$

There are, however, many other metrics apart from the accuracy one that play a vital role in other classification problems: recall, precision, F1 score, true positive rate and so on. Here we believe that the accuracy metric is enough to draw conclusions about how the image inpainting is improving the

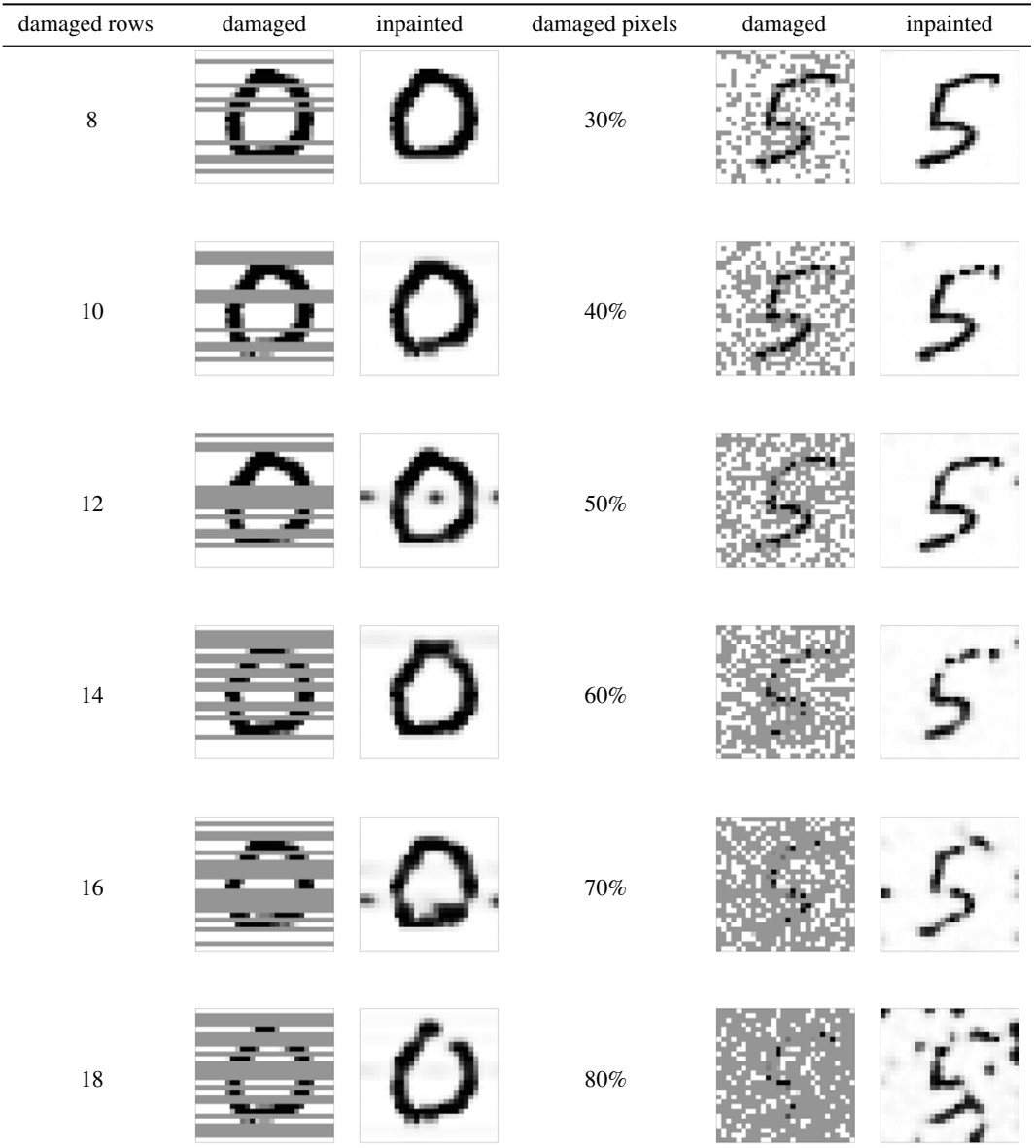

| damaged rows | damaged | inpainted | damaged pixels | damaged | inpainted |
|---|---|---|---|---|---|
| 8 | | | 30% | | |
| 10 | | | 40% | | |
| 12 | | | 50% | | |
| 14 | | | 60% | | |
| 16 | | | 70% | | |
| 18 | | | 80% | | |

**Figure 5.** Examples of image inpainting for random damage in whole horizontal rows and in individual pixels. The column entitled 'damaged rows' marks the number of rows randomly selected for damage in the 28 × 28 images. The column entitled 'damaged pixels' marks the percentage of randomly damaged pixels over the whole image. For higher levels of damage intensity, the inpaintings lose more information.

predictions with respect to the damage images. This is due to the fact that the MNIST dataset is a balanced dataset, where there is generally no preference between false positives and false negatives.

The measure of improvement between the batch of samples with image inpainting and the damaged ones without it is computed as

$$\text{improvement} \equiv \frac{\text{accuracy}_{\text{with CH filter}} - \text{accuracy}_{\text{without CH filter}}}{\text{accuracy}_{\text{without CH filter}}}, \tag{3.1}$$

and it basically represents the percentage of improvement that results from adding the CH filter to the prediction process.

The results for all types of damage under consideration are displayed in tables 3–5. In table 3, we gather the prediction accuracies for the four customized damages displayed in figure 4, as well as the prediction accuracy for the unmodified MNIST test set, which for our neural network architecture is 0.97. We observe that for the types of more intense customized damage B and C the accuracy prediction for the damaged images without CH filter drops to 0.71 and 0.64, respectively. By applying

**Table 3.** Accuracy for the test dataset of MNIST without and with the CH filter, for the customized damage in figure 4. The improvement is computed following [21].

| customized damage | without CH filter | with CH filter | improvement |
|---|---|---|---|
| A | 0.84 | 0.96 | 14% |
| B | 0.71 | 0.93 | 31% |
| C | 0.64 | 0.82 | 28% |
| D | 0.90 | 0.96 | 7% |
| initial test images | — | 0.97 | — |

**Table 4.** Accuracy for the test dataset of MNIST without and with the CH filter, for the case of random damage in rows. The improvement is computed following [21].

| damaged rows | without CH filter | with CH filter | improvement |
|---|---|---|---|
| 6 | 0.89 | 0.96 | 8% |
| 8 | 0.82 | 0.93 | 13% |
| 10 | 0.73 | 0.91 | 25% |
| 12 | 0.66 | 0.87 | 32% |
| 14 | 0.6 | 0.87 | 45% |
| 16 | 0.55 | 0.81 | 47% |
| 18 | 0.47 | 0.68 | 45% |
| 20 | 0.40 | 0.48 | 20% |
| 22 | 0.39 | 0.45 | 15% |
| 24 | 0.33 | 0.26 | −21% |
| 26 | 0.20 | 0.12 | −40% |

**Table 5.** Accuracy for the test dataset of MNIST without and with the CH filter, for the case of random damage in pixels. The improvement is computed following [21].

| damaged pixels | without CH filter | with CH filter | improvement |
|---|---|---|---|
| 30% | 0.93 | 0.93 | 0% |
| 40% | 0.96 | 0.96 | 0% |
| 50% | 0.91 | 0.95 | 4% |
| 60% | 0.8 | 0.94 | 18% |
| 70% | 0.75 | 0.93 | 24% |
| 80% | 0.55 | 0.8 | 45% |
| 90% | 0.39 | 0.46 | 18% |
| 92% | 0.32 | 0.37 | 16% |
| 94% | 0.33 | 0.34 | 3% |
| 96% | 0.20 | 0.23 | 15% |

the filter we find that the accuracy predictions can significantly escalate to 0.93 and 0.82, leading to improvements of 31% and 28%, respectively. The other two types of customized damage, A and D, are not as pervasive as B and C, and as a result the accuracy predictions are high even without applying the CH filter.

In tables 4 and 5, we test the accuracies for random damage with various levels of intensities. The objective here is to analyse how the CH filter responds when the damage occupies more and more

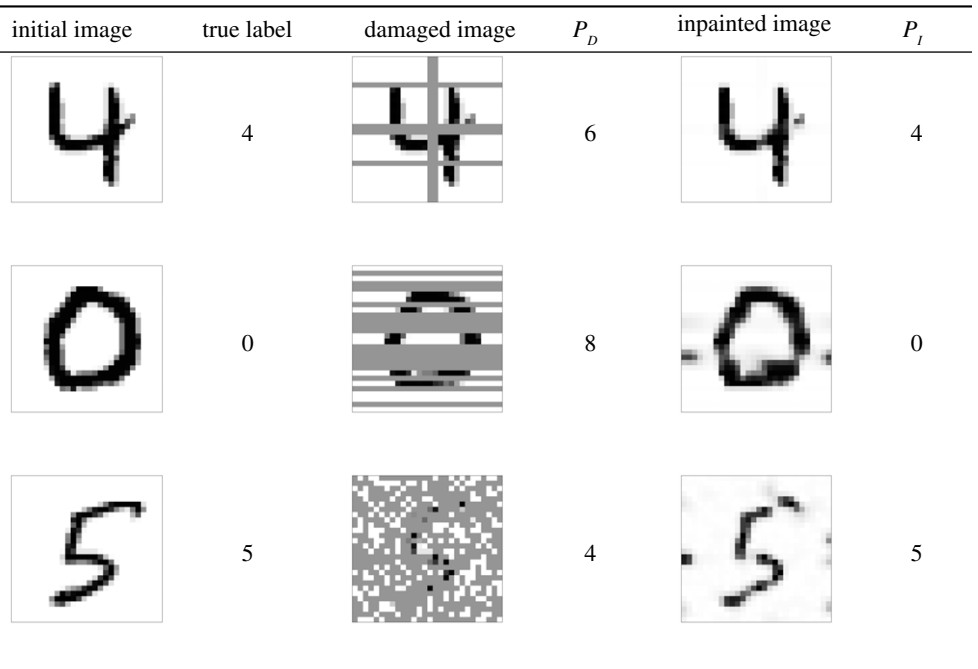

| initial image | true label | damaged image | $P_D$ | inpainted image | $P_I$ |
|---|---|---|---|---|---|
| | 4 | | 6 | | 4 |
| | 0 | | 8 | | 0 |
| | 5 | | 4 | | 5 |

**Figure 6.** Particular examples of label predictions for MNIST samples with various types of damage: customized damage C, random damage in 16 rows and random damage in 70% of pixels. The label of the damaged image is wrongly predicted, while the label for the inpainted image is correctly predicted. $P_D$ and $P_I$ represent the label predictions of the damaged and inpainted images, respectively.

space in the images, both for the case of rows or pixels, as displayed in figure 5. In table 4, we show the accuracies for a range of damaged rows between 6 and 26, bearing in mind that the dimensions of the MNIST images are $28 \times 28$. We observe that for low numbers of damaged rows, the accuracy prediction even without the CH filter is high and it does not improve significantly by adding the filter. But then the improvement surges until reaching a maximum value of 47% for 16 random damaged rows, where the prediction without CH filter is 0.55 and with CH filter 0.81. For larger numbers of damaged rows the accuracies drastically drop due to the large amount of information lost, and not even the image inpainting process is able to achieve decent accuracies. In the limit of the damaged number of rows tending to 28, we observe that the accuracies are close to the ones of a dummy classifier with one out of ten chances of rightly guessing the label. In this limit, there is no difference between adding the CH filter or not, and it turns out that the improvements are even negative.

We observe a similar pattern for the case of random damaged pixels in table 5. For low percentages of damaged pixels, the improvement of adding the CH filter is negligible and the accuracies with and without the filter are quite high. As we increase the percentage of damaged pixels the improvement escalates until it reaches 45% for a scenario with 80% of the pixels randomly damaged. For this case, the accuracy prediction without the filter is just 0.55, but thanks to the filter it significantly increases to a decent value of 0.8. For larger percentage of pixels, the improvement and accuracies drop, and in the limit towards 100% of damaged pixels we get close to the accuracy of a dummy classifier. This is due to the large loss of information that the original images have suffered.

In figure 6, we depict some specific examples for which the label is only predicted correctly after applying the CH filter to the damaged image. These are just some particular samples out of the 10 000 images contained in the test dataset of MNIST, and for some of them the opposite effect can occur: that the label is correctly predicted for the damaged image but following the inpainting process it is predicted incorrectly. This only occurs for really severe damages, where there is no difference between adding the CH filter or not and the accuracies are close to the ones of a dummy classifier guessing randomly. However, we have shown in tables 4 and 5 that, overall, the CH filter increases the global accuracy for images with low to moderate damage, and consequently we expect that scenarios such as the ones displayed in figure 6 are much more common than the opposite ones.

In figure 6, we also find that some greyscale pixels remain in the inpainted image after applying the modified CH filter. This is due to the fact that the finite-volume scheme has not run for long enough to

turn those intensities into the stable white ($\phi = -1$) or black ($\phi = 1$) phase field. This is prone to happen for images with high levels of damage, and one should try to let the simulation run for as long as possible to avoid possible misclassification issues due to those greyscale regions.

# 4. Discussion and conclusion

We have quantified the prediction improvement of employing a CH image inpainting filter to restore damaged images which are then passed into a neural network. We combined a finite-volume scheme with a neural network for pattern recognition to develop an integrated algorithm summing up the process of adding damage to the images and then predicting their label. Our results for the MNIST dataset suggest that, in general, the accuracy is improved for a wide range of low to moderate damages, while for some particular cases we reach improvements of up to 50%. We also provide the image inpainting outcome of multiple damage scenarios and the benefits of adding the CH filter to predict the label of the image are easily visible.

We believe that our results employing the MNIST dataset lay the foundations towards the application of image inpainting in more complex datasets. We have demonstrated the benefit of combining the fields of image inpainting with machine learning, and we believe that many applications can take advantage of it. For instance, there are applications such as medical images from MRI or satellite observations where typically there is some inherent noise or damage involved and where there may be potential to employ tools from machine learning as was done here.

MNIST is one of the most well-known and convenient benchmark datasets. It is possible that applying our methodology to increasingly complex datasets might bring about new challenges. At the same time, we have relied on two main assumptions about the damage and the images: the first one is that the images are binary, leading to the standard CH potential in (1.4) that has only two wells (i.e. one for each of the two colours); the second one is that the damage is not blind, meaning that the location of the damage is known. Performing image inpainting without these two assumptions becomes substantially more involved as already pointed out in [7,23,26]. We will be exploring these and related questions in future works.

Data accessibility. Data and relevant code for this research work are stored in GitHub and have been archived within the Zenodo repository. See https://github.com/sergiopperez/Image_Inpainting and doi:10.5281/zenodo.4616628.

Competing interests. We declare we have no competing interests.

Funding. S.P.P. acknowledges financial support from the IC President's PhD Scholarship. J.A.C. was partially supported by the EPSRC through grant no. EP/P031587 and the ERC through Advanced grant no. 883363. S.K. was partially supported by the EPSRC through grant no. EP/L020564 and the ERC through Advanced grant no. 247031.

Acknowledgements. We are indebted to P. Yatsyshin and A. Russo from the Chemical Engineering Department of Imperial College (IC) for numerous stimulating discussions on machine learning and image inpainting.

# Appendix A. Sensitivity analysis of the parameters $\epsilon_1$, $\epsilon_2$ and $\lambda$

Adequately tuning the two values of $\epsilon_1$, $\epsilon_2$ and $\lambda$, is vital to complete a successful image inpainting. These values have to be chosen empirically and depend on both the dataset and type of damage. Here we provide further details about finding an adequate combination of these parameters for images of MNIST.

On the one hand, the parameter $\epsilon$ is related to the interphase thickness of the solution of the modified CH equation in (1.6). The larger the $\epsilon$, the more diffused the resulting image. However, a large $\epsilon$ allows for a large-scale topological reconnection of shapes, which is convenient when the damage breaks a connected colour phase. The strategy proposed in [21] and followed here, consists of employing a two-stage methodology where a large $\epsilon$ is employed first (denoted as $\epsilon_1$), followed by a smaller $\epsilon$ (denoted as $\epsilon_2$) in order to sharpen the final image. In figure 7, we depict some of the image-inpainted images resulting from various combinations of $\epsilon_1$ and $\epsilon_2$. The original image with damage is shown in figure 7a, and the optimal combination of $\epsilon_1$ and $\epsilon_2$ following the values in table 2 is depicted in figure 7b. For this example, the optimal parameters are $\epsilon_1 = 1.5$ and $\epsilon_2 = 0.5$, and depend on the mesh size that in this case is $\Delta x = \Delta y = 1$. A larger mesh size requires a larger value of $\epsilon$ to produce the same outcome, implying that both parameters are interconnected and have to be of the same order. In order to evaluate the impact of $\epsilon_1$ and $\epsilon_2$, we run two experiments: in the first one we choose both of them with the small value of $\epsilon_1 = \epsilon_2 = 0.5$, and in the second one both parameters take the large value $\epsilon_1 = \epsilon_2 = 1.5$. The outcome with the small value is depicted in figure 7c, and clearly there is not an efficient large-scale topological reconnection of the white regions, resulting in some damage left in the image. The outcome with the large value is depicted in figure 7d, and the drawback here is that the image is too diffused in comparison to the optimal combination in figure 7b.

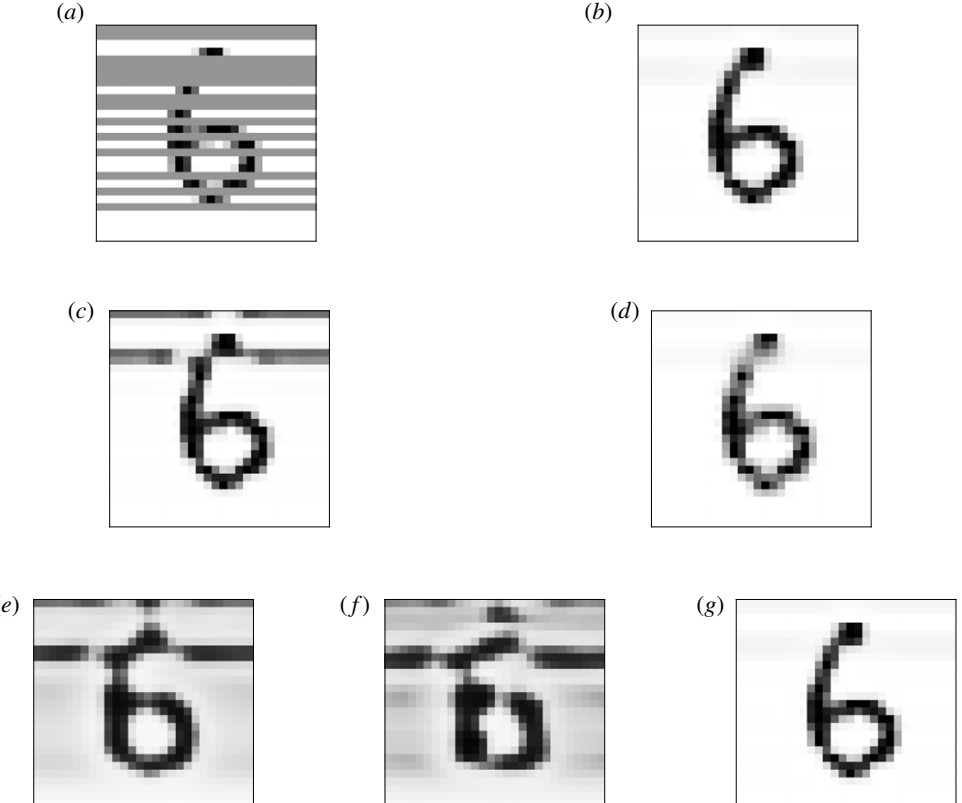

**Figure 7.** Sensitivity analysis for different values of $\epsilon_1$, $\epsilon_2$ and $\lambda$. (*a*) original image with damage; (*b*) optimal choice of parameters; (*c*) short value of $\epsilon_1$ and $\epsilon_2$; (*d*) large value of $\epsilon_1$ and $\epsilon_2$; (*e*) really small value of $\lambda$; (*f*) small value of $\lambda$; (*g*) large value of $\lambda$. (*a*) Original image with damage, (*b*) $\epsilon_1 = 1.5$, $\varepsilon_2 = 0.5$ and $\lambda = 1000$, (*c*) $\epsilon_1 = 0.5$, $\epsilon_2 = 0.5$ and $\lambda = 1000$, (*d*) $\epsilon_1 = 1.5$, $\epsilon_2 = 1.5$ and $\lambda = 1000$, (*e*) $\epsilon_1 = 1.5$, $\epsilon_2 = 0.5$ and $\lambda = 0.01$, (*f*) $\epsilon_1 = 1.5$, $\epsilon_2 = 0.5$ and $\lambda = 0.1$, (*g*) $\epsilon_1 = 1.5$, $\epsilon_2 = 0.5$ and $\lambda = 10\,000$.

On the other hand, the parameter $\lambda$ in the finite-volume scheme in (2.4) ensures that the undamaged pixels are not modified during the image inpainting. The choice of $\lambda$ has to be sufficiently large to counterbalance the fluxes of the scheme and act as a penalty term that keeps the undamaged pixels invariant. However, $\lambda$ cannot be too large since otherwise the finite-volume scheme becomes a singularly perturbed problem, and the convergence of the implicit scheme is deteriorated. The value of $\lambda$ is related to the time step $\Delta t$, and greater values of $\lambda$ are possible if $\Delta t$ is refined. In our case, with the choice of $\Delta t = 0.1$, our finite-volume scheme does not yield any result and breaks down during the simulation for values of $\lambda \notin [1, 10\,000]$. To evaluate the impact of $\lambda$ we run three experiments: in the first one we choose the really small value $\lambda = 0.01$, in the second the small $\lambda = 0.1$ and in the third the large value $\lambda = 10\,000$. The outcomes with the two small values are depicted in figure 7*e*,*f*, and we observe that the undamaged pixels have been altered and the overall appearance is not clean. The outcome with the large value is depicted in figure 7*g*, and the image is comparable to the optimal one in figure 7*b*. There is a threshold-$\lambda$ value above which there is no convergence of the implicit finite-volume scheme. For the particular image, we found this threshold can be approximated by $\lambda = 10\,000$.

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
