## [Peer Review File · Royal Society Open Science]

Review History

RSOS-201294.R0 (Original submission)

Review form: Reviewer 1

Is the manuscript scientifically sound in its present form?

Yes

Are the interpretations and conclusions justified by the results?

Yes

Is the language acceptable?

Yes

Do you have any ethical concerns with this paper?

No

Have you any concerns about statistical analyses in this paper?

No

Recommendation?

Accept with minor revision (please list in comments)

Comments to the Author(s)

See attached report (Appendix A).

Decision letter (RSOS-201294.R0)

Dear Mr P. Perez

The Editors assigned to your paper RSOS-201294 "Enhancement of damaged-image prediction through Cahn-Hilliard Image Inpainting" have now received comments from reviewers and would like you to revise the paper in accordance with the reviewer comments and any comments from the Editors. Please note this decision does not guarantee eventual acceptance.

Please submit your revised manuscript and required files (see below) no later than 21 days from today's (ie 08-Feb-2021) date. Note: the ScholarOne system will 'lock' if submission of the revision is attempted 21 or more days after the deadline. If you do not think you will be able to meet this deadline please contact the editorial office immediately.

Best regards,

on behalf of Professor Anotida Madzvamuse (Associate Editor) and Mark Chaplain (Subject Editor)

Associate Editor Comments to Author (Professor Anotida Madzvamuse):

1. A clear description of the terms introduced into the CH model must be described fully, these lead to the modified CH. Their physical interpretation must be provided.
2. It is not clear to this reviewer what is the definition of MNIST? Also why is this data important to study and use as a benchmark?
3. Is the accuracy independent of the dataset?
4. ϵ is not defined in ϕ (epsilon) just under equation (1).
5. The statement "As already alluded to above .." should be rephrased
6. The operators F and G in (8) are not defined.
7. How is ϕ_{ij}^0 chosen? Does the choice of the initial phase field function impact on the accuracy?
8. The inpainted domain D is not defined.
9. In (10), what is u and where does it come from? The u are referred as velocities, for which model?
10. Similarly what is v ?
11. Next, you introduce $\xi_{i,j}$, what are these. I see no model with any of these variables. This section is extremely confusing for the reader.
12. In (11), the timestep $(n+1)$ is reflected on the left-hand side but then disappears on the right hand side. Please correct.
13. Furthermore, two functions are introduced, $H_c(\rho)$ and $H_e(\rho)$, what are these functions or operators? What is ρ ?
14. Now ξ is defined in (13), how can this be if it was stated a priori without any notion nor definition? First, define the variables before they are used.
15. In (14) one definition for the discrete Laplace operator is sufficient at one time level.
16. In (15), boundary flux conditions are now defined in abstract operators, yet these are incorrectly defined. Please define the boundary conditions fully explicitly.
17. The two-step method in Section 2.2 is not well described. The authors state: "It basically consists of dividing the image inpainting in two subsequent stages, ...". This is not physically meaningful, to divide an image into two steps? The domain is discretised using two different discretisation.... I would assume.
18. It is not clear if the choices of ϵ and λ are done by trial and error or there is a systematic way of selecting these.
19. Also, how independent is the accuracy of ϵ_1 and ϵ_2 ?
20. Does the choice of $\Delta x = 1 = \Delta y$ satisfy the CFL condition?
21. Does the choice of the filter matter? Can similar accuracy be obtained with a different filter?
22. Page 5, item 2, what is the domain of definition for x ?
23. Can the authors do a sensitivity analysis of the parameters? λ seem to belong a very large range while ϵ seems reasonable. How sensitive are the results to these parameters?
24. How sensitive are the results to mesh and timestep refinements?

Reviewer comments to Author:

Reviewer: 1

Comments to the Author(s)

See attached report

===PREPARING YOUR MANUSCRIPT===

===PREPARING YOUR REVISION IN SCHOLARONE===

Author's Response to Decision Letter for (RSOS-201294.R0)

See Appendices B & C.

Decision letter (RSOS-201294.R1)

Dear Mr P. Perez,

It is a pleasure to accept your manuscript entitled "Enhancement of damaged-image prediction through Cahn-Hilliard Image Inpainting" in its current form for publication in Royal Society Open Science. The comments of the reviewer(s) who reviewed your manuscript are included at the foot of this letter.

on behalf of Professor Anotida Madzvamuse (Associate Editor) and Mark Chaplain (Subject Editor)
openscience@royalsociety.org

Associate Editor Comments to Author (Professor Anotida Madzvamuse):
Comments to the Author:

The authors have substantially addressed comments raised by the reviewers. I am happy to recommend acceptance of the manuscript.

Appendix A

Review of “Enhancement of damaged-image prediction through Cahn-Hilliard Image Inpainting”

In this contribution the authors proposed to use an image inpainting filter to aid a neural-network-based classification of damaged images of digits. The inpainting is done via a modified Cahn–Hilliard equation implemented numerically with a finite volume scheme. By integrating with a 2-layer neural network, the resulting inpainting-aided classification is tested on the MNIST dataset damaged by two types of occlusions. The authors then conducted various experiments and compared the classification accuracy with and without the proposed Cahn–Hilliard filter for various levels of damage.

The methodology proposed in the contribution is sound, and to the best of my knowledge, while the use of neural networks for image classification and the use of Cahn–Hilliard equation for inpainting have been around for at least a decade, this combination proposed by the authors is novel. The authors provided detailed explanations, complemented with tables of data, regarding parameter choices and the advantages of employing the Cahn–Hilliard inpainting as a pre-processing step.

In my view, it seems the main goal of this Cahn–Hilliard pre-processing filter is not about obtaining a high quality reconstruction of the damaged image (as originally intended in [11]), but rather to “fix” the damage just enough for the subsequent neural network to correctly classify the image. This is evident from Table 5 where the Cahn–Hilliard filter made no obvious improvements for low levels of damage, but offers significant improvements for medium levels of damage. In light of this viewpoint, one may argue that a different PDE can be used instead of the Cahn–Hilliard equation, such as the 2nd order Allen–Cahn equation (modified with the same inpainting term) - which may be even faster to solve, or even replacing the Cahn–Hilliard filter with another neural network. Nevertheless, I believe that present contribution deserves some merit in approaching the digital inpainting subproblem with a mathematical methodology consistent with the principles of inpainting, as opposed to leaving everything to a rather opaque neural-network framework.

Overall, the manuscript is highly readable and easy to understand. The results are interesting and the paper is a good fit for the journal. I recommend publication provided the criticism and question below are addressed.

Minor criticism:

- Emphasis on the advantages of the finite-volume scheme is somewhat lacking. While the authors mentioned properties such as discrete energy decay, unconditionally stable, and reduction in computational costs - proven probably (since I was not able to access) in [6] for the original Cahn–Hilliard equation, it is unknown just how much can be transferred to this modified Cahn–Hilliard equation (at least not evident from the corresponding paragraphs). In particular, the modified Cahn–Hilliard equation does not admit a gradient flow structure with the Ginzburg–Landau energy.

Moreover, it is not entirely convincing why practitioners might consider to use this finite-volume scheme as opposed to well-preconditioned finite element/finite difference methods or FFT-based spectral methods. Perhaps the advantages are less evident here due to the structure of the MNIST data set of 28 by 28 pixels, and as the authors have highlighted, the real advantages might materialise in the higher-dimensional setting.

Question:

- In the paragraph above Figure 6, the authors mentioned there are examples where the label is correctly predicted for the damaged image but incorrectly predicted for the inpainted image. Did the authors identify what went wrong for these examples?

I noticed several artefacts or regions of grayscale away from the digit region appearing in the inpainted results in Figure 5. Perhaps these artefacts might be responsible for some misclassifications of the inpainted images? And if so, would some type of ad-hoc thresholding for the inpainted image, leading to a true binary image as inputs for the classification step, “fix” some of these misclassified examples?

Appendix B

Enhancement of damaged-image prediction through Cahn-Hilliard Image Inpainting

REF. # RSOS-201294

José A. Carrillo, Serafim Kalliadasis, Fuyue Liang and Sergio P. Perez,
RESPONSE TO COMMENTS BY ASSOCIATE EDITOR

We are grateful to the Associate Editor (Prof. Anotida Madzvamuse, U. Sussex) for carefully reading our manuscript and for bringing up a number of insightful comments which we feel have helped us improve its quality.

We proceed by addressing in turn each of the Associate Editor's comments, reproduced below in italics. For the sake of clarity, changes-modifications in the revised manuscript in the response to the Associate Editor are highlighted in red (while changes-modifications in response to Reviewer 1 are highlighted in blue).

1. A clear description of the terms introduced into the CH model must be described fully, these lead to the modified CH. Their physical interpretation must be provided.

Good point, thank you. We have added the following paragraph to the introduction, in p. 2:

"There is a clear physical interpretation of the terms forming the free energy in (2): on the one hand, the potential $H(\phi)$ contains the equilibrium information of the system, and each of the minima of the choice (4) correspond to an equilibrium phase; on the other hand, the gradient term in (2) accounts for the cost of spatial inhomogeneities, resulting in a surface tension between different phases. As a result, the equilibrium state balances the tendency of separation from the hydrophobic potential $H(\phi)$ and the tendency of mixing from the hydrophilic gradient term. The mobility term impacts the rate of phase separation and the coarsening process, and some works employ nonconstant degenerate mobilities that cancel when the phase field corresponds to one of the two wells in (4) (see e.g. [3, 6, 46] for further discussions of the physical significance of the various terms of the CH equation)."

2. It is not clear to this reviewer what is the definition of MNIST? Also why is this data important to study and use as a benchmark?

Nice suggestion. We have clarified this both in the abstract and in the introduction. In the abstract we added the following sentence:

"The benchmark dataset employed here is MNIST, which consists of binary images of handwritten digits and is a standard dataset to validate image-processing methodologies."

We have also added the following text on p. 3 of the Introduction:

"The MNIST database (Modified National Institute of Standards and Technology database) is a standard dataset to validate image-processing methodologies, and it contains binary handwritten images of numbers from zero to nine. We choose this dataset since the modified CH equation (6) is applicable to binary images such as the ones in the MNIST dataset. The extension of this work to non-binary images is also relevant and will be explored elsewhere."

3. Is the accuracy independent of the dataset?

In this work we focus specifically on the MNIST dataset, with the objective of validating a proof of concept for an image-inpainting methodology that can be extended to more complex datasets in the future. Consequently, all the accuracy results showed in our manuscript refer to the MNIST dataset exclusively.

Following your remark we have added a disclaimer on p. 3 of the Introduction:

“This increase in accuracy is obtained by applying our image-inpainting methodology to the MNIST dataset exclusively, and as a disclaimer we remark that other datasets or types of damage may result in different accuracy increases.”

4. ϵ is not defined in $\phi(\epsilon)$ just under equation (1).

Well spotted. It is true that ϵ is only defined later when introducing the modified CH equation. We have now added the following sentence in the paragraph following Eq. (1):

“with ϵ being a positive parameter related to the interface thickness coming from the derivation of diffuse-interface models [19].”

In this sentence we have cited new reference [19]:

M. Cates, J.-C. Desplat, P. Stansell, A. Wagner, K. Stratford, R. Adhikari, and I. Pagonabarraga, Physical and computational scaling issues in lattice Boltzmann simulations of binary fluid mixtures, Philosophical Transactions of the Royal Society A: Mathematical, Physical and Engineering Sciences, 363 (2005), pp. 1917–1935.

5. The statement “As already alluded to above ..” should be rephrased

Good point. We have modified the sentence as follows:

“One of the main advantages of employing the CH equation for image inpainting is the myriad of fast and reliable numerical methods available for its solution.”

6. The operators F and G in (8) are not defined.

Good point also. We have added a clarification about F and G below Eq. (10) (Equation (8) in the original manuscript):

“with $F_{i+1/2,j}^{n+1}$ and $G_{i,j+1/2}^{n+1}$ being flux approximations at the boundaries.”

We have also added more comments on the definition of F and G below Eq. (11) (Equation (9) in the original manuscript):

“The flux terms $F_{i+1/2,j}^{n+1}$ and $G_{i,j+1/2}^{n+1}$ are the discrete approximations of the velocity components v and w at the cell interfaces $(x_{i+1/2}, y_j)$ and $(x_i, y_{j+1/2})$, respectively. Such approximations follow an upwind and implicit approach inspired by the works in Refs. [5, 6, 12, 17], satisfying

$$F_{i+\frac{1}{2},j}^{n+1} = \left(v_{i+1/2,j}^{n+1}\right)^+ + \left(v_{i+1/2,j}^{n+1}\right)^-,$$
$$G_{i,j+\frac{1}{2}}^{n+1} = \left(w_{i,j+1/2}^{n+1}\right)^+ + \left(w_{i,j+1/2}^{n+1}\right)^-.$$

”

7. How is $\phi_{i,j}^0$ chosen? Does the choice of the initial phase field function impact on the accuracy?

Good point once again. $\phi_{i,j}^0$ is actually the original image containing the damage. This means that $\phi_{i,j}^0$, rather than being chosen, is directly given from the dataset. $\phi_{i,j}^0$ corresponds to the pixel intensities of the images in that dataset.

The reason for including this term in Eq. (11) is to avoid modifying the cells that are outside the inpainting domain. More specifically, in the expression $\lambda_{i,j}(\phi_{i,j}^0 - \phi_{i,j}^n)$ we choose $\lambda_{i,j}$ to be nonzero only outside the inpainting domain. As a result, the cells outside the inpainting domain are penalized if they deviate from their original values.

But we can see the confusion here, and we have now clarified the text in relation to $\phi_{i,j}^0$:

“The time step is denoted as Δt . In each of the cells we define the cell average $\phi_{i,j}^n$ at time $t = n\Delta t$ as

$$\phi_{i,j}^n = \frac{1}{\Delta x \Delta y} \int_{C_{i,j}} \phi(x, y, t = n\Delta t) dx dy,$$

where $\phi_{i,j}^0$ is the phase field at $t = 0$, which corresponds to the normalized pixel intensities of the initial damaged image to be inpainted.”

8. The inpainted domain D is not defined.

We have added this clarification about the inpainted domain below Eq. (12) (Equation (9) in the original manuscript):

“with D being the inpainted domain where the image damage is located. The inpainted domain has to be determined beforehand for each image, and is formed by those finite-volume cells to be repaired during the image inpainting.”

9. In (10), what is u and where does it come from? The u are referred as velocities, for which model?

10. Similarly what is v ?

11. Next, you introduce $\xi_{i,j}$, what are these. I see no model with any of these variables. This section is extremely confusing for the reader.

14. Now ξ is defined in (13), how can this be if it was stated a priori without any notion nor definition? First, define the variables before they are used.

Thanks for giving us the opportunity to rewrite and clarify the notation of all these variables. We have now firstly introduced the continuum versions of the variables \mathbf{u} , v , w and ξ at the start of subsection 2.1:

“For simplicity, let's rewrite (6) in 2D by introducing $\mathbf{u} = (v, w)$ as the physical velocity term and ξ as the variation of the free energy with respect to the density,

$$\frac{\partial \phi(x, y, t)}{\partial t} = -\nabla \cdot \mathbf{u} + \lambda(x, y) (\phi(x, y, t = 0) - \phi(x, y, t)),$$

where

$$\xi = \frac{\delta \mathcal{F}[\phi]}{\delta \phi} = \epsilon^2 \nabla^2 \phi - H'(\phi), \quad \mathbf{u} = \nabla \xi, \quad v = \frac{\partial \xi}{\partial x}, \quad \text{and} \quad w = \frac{\partial \xi}{\partial y}.$$

..

Subsequently, in the rest of subsection 2.1 we keep the same notation when introducing the discretized form of the variables u , v , w and ξ . We believe that this notation is now clear to the reader.

12. In (11), the timestep $(n+1)$ is reflected on the left-hand side but then disappears on the right hand side. Please correct.

Well spotted, this is now corrected:

$$v_{i+\frac{1}{2},j}^{n+1} = -\frac{\xi_{i+1,j}^{n+1} - \xi_{i,j}^{n+1}}{\Delta x}, \quad w_{i,j+\frac{1}{2}}^{n+1} = -\frac{\xi_{i,j+1}^{n+1} - \xi_{i,j}^{n+1}}{\Delta y},$$

13. Furthermore, two functions are introduced, $H_c(\rho)$ and $H_e(\rho)$, what are these functions or operators? What is ρ ?

Good point also, thank you. We have introduced the following clarification on p. 5:

“where the potential $H(\phi)$ in (4) is decomposed into two convex functions, $H_c(\phi)$ and $H_e(\phi)$,

$$H(\phi) = H_c(\phi) - H_e(\phi) = \frac{\phi^4 + 1}{4} - \frac{\phi^2}{2}.$$

”

We note $H_c(\rho)$ and $H_e(\rho)$ are actually $H_c(\phi)$ and $H_e(\phi)$. The inclusion of ρ in the text was a typo, we apologise for the remiss.

15. In (14) one definition for the discrete Laplace operator is sufficient at one time level.

That is 100% true. We have now kept only one definition of the Laplacian:

“The discrete two-dimensional approximation of the Laplacian $(\Delta\phi)_{i,j}^n$ is chosen to satisfy the second-order form

$$(\Delta\phi)_{i,j}^n := \frac{\phi_{i+1,j}^n - 2\phi_{i,j}^n + \phi_{i-1,j}^n}{\Delta x^2} + \frac{\phi_{i,j+1}^n - 2\phi_{i,j}^n + \phi_{i,j-1}^n}{\Delta y^2}.$$

”

16. In (15), boundary flux conditions are now defined in abstract operators, yet these are incorrectly defined. Please define the boundary conditions fully explicitly.

Another good point. We now specify the boundary conditions at an early stage, i.e. the Introduction:

“The boundary conditions imposed for the CH equation in (1) are no-flux for both the phase field and for the variation of the free energy,

$$\epsilon^2 \nabla\phi \cdot \mathbf{n} = 0, \quad M(\phi) \nabla\xi \cdot \mathbf{n} = 0,$$

where \mathbf{n} is defined in the normal direction to and into the wall.”

Subsequently, in subsection 2.1, we have provided more detail about the numerical implementation of the boundary conditions:

“The modified CH equation in (6) employs the no-flux boundary conditions defined in (5). The numerical implementation of the boundary conditions follows from

$$\begin{aligned}
 \phi_{i-1,j}^n &= \phi_{i,j}^n \text{ for } i = 1, \forall j; & \phi_{i+1,j}^n &= \phi_{i,j}^n \text{ for } i = N, \forall j; \\
 \phi_{i,j-1}^n &= \phi_{i,j}^n \text{ for } j = 1, \forall i; & \phi_{i,j+1}^n &= \phi_{i,j}^n \text{ for } j = N, \forall i; \\
 F_{i-\frac{1}{2},j}^{n+1} &= 0 \text{ for } i = 1, \forall j; & F_{i+\frac{1}{2},j}^{n+1} &= 0 \text{ for } i = N, \forall j; \\
 G_{i,j-\frac{1}{2}}^{n+1} &= 0 \text{ for } j = 1, \forall i; & G_{i,j+\frac{1}{2}}^{n+1} &= 0 \text{ for } j = N, \forall i.
 \end{aligned}$$

”

17. The two-step method in Section 2.2 is not well described. The authors state: “It basically consists of dividing the image inpainting in two subsequent stages, ...”. This is not physically meaningful, to divide an image into two steps? The domain is discretised using two different discretisation.... I would assume.

We can see the confusion here. By “dividing the image inpainting” we mean the process of running the finite-volume scheme in subsection 2.1. This just means that we run it twice: first, with a large ϵ , and second, with a smaller ϵ .

Our sentence is clearly confusing, and we have now modified it as follows:

“The two-step method followed here was firstly proposed in [11]. It basically consists of applying the finite-volume scheme in subsection 2.1 twice with different values of the parameter ϵ .”

18. It is not clear if the choices of ϵ and λ are done by trial and error or there is a systematic way of selecting these.

19. Also, how independent is the accuracy of ϵ_1 and ϵ_2 ?

23. Can the authors do a sensitivity analysis of the parameters? λ seem to belong a very large range while epsilon seems reasonable. How sensitive are the results to these parameters?

24. How sensitive are the results to mesh and timestep refinements?

Nice remarks. We fully acknowledge that there is room to better explain the choices of all these parameters. We have now added a dedicated Appendix aimed at clarifying the choice of the parameters ϵ and λ and their relation to the mesh size Δx and time step Δt . In the Appendix we also undertake a sensitivity analysis of the parameters ϵ and λ , which is supported by new Figure 7 depicting the outcome of various combinations of ϵ_1 , ϵ_2 and λ .

For completeness, we reproduce here the content of the Appendix and new Fig. 7:

“Adequately tuning the two values of ϵ_1 , ϵ_2 and λ , is vital to complete a successful image inpainting. These values have to be chosen empirically and depend on the dataset and type of damage. Here we provide further details about finding an adequate combination of these parameters for images of MNIST.

On the one hand, the parameter ϵ is related to the interphase thickness of the solution of the modified CH equation in (6). The larger the ϵ , the more diffused the resulting image. However, a large ϵ allows for a large-scale topological reconnection of shapes, which is convenient when the damage breaks a connected color phase. The strategy proposed in

[11] and followed here, consists of employing a two-stage methodology where a large ϵ is employed first (denoted as ϵ_1), followed by a smaller ϵ (denoted as ϵ_2) in order to sharpen the final image. In Fig. 7 we depict some of the image-inpainted images resulting from various combinations of ϵ_1 and ϵ_2 . The original image with damage is shown in Fig. 7a, and the optimal combination of ϵ_1 and ϵ_2 following the values in Table 2 is depicted in Fig. 7b. For this example the optimal parameters are $\epsilon_1 = 1.5$ and $\epsilon_2 = 0.5$, and depend on the mesh size that in this case is $\Delta x = \Delta y = 1$. A larger mesh size requires a larger value of ϵ to produce the same outcome, implying that both parameters are interconnected and have to be of the same order. In order to evaluate the impact of ϵ_1 and ϵ_2 we run two experiments: in the first one we choose both of them with the small value of $\epsilon_1 = \epsilon_2 = 0.5$, and in the second both parameters take the large value $\epsilon_1 = \epsilon_2 = 1.5$. The outcome with the small value is depicted in Fig.7c, and clearly there is not an efficient large-scale topological reconnection of the white regions, resulting in some damage left in the image. The outcome with the large value is depicted in Figure 7d, and the drawback here is that the image is too diffused in comparison to the optimal combination in Figure 7b.

On the other hand, the parameter λ in the finite-volume scheme in (6) ensures that the undamaged pixels are not modified during the image inpainting. The choice of λ has to be sufficiently large to counterbalance the fluxes of the scheme and act as a penalty term that keeps the undamaged pixels invariant. However, λ cannot be too large since otherwise the finite-volume scheme becomes a singularly perturbed problem, and the convergence of the implicit scheme is deteriorated. The value of λ is related to the time step Δt , and greater values of λ are possible if Δt is refined. In our case, with the choice of $\Delta t = 0.1$, our finite-volume scheme does not yield any result and breaks down during the simulation for values of $\lambda \notin [1, 10000]$. To evaluate the impact of λ we run three experiments: in the first one we choose the really small value $\lambda = 0.01$, in the second the small $\lambda = 0.1$ and in the third the large value $\lambda = 10000$. The outcomes with the two small values are depicted in Figs 7e and 7f, and we observe that the undamaged pixels have been altered and the overall appearance is not clean. The outcome with the large value is depicted in Fig. 7g, and the image is comparable to the optimal one in Fig. 7b. There is a threshold- λ value above which there is no convergence of the implicit finite-volume scheme. For the particular image we found this threshold can be approximated by $\lambda = 10000$."

20. Does the choice of $\Delta x = 1 = \Delta y$ satisfy the CFL condition?

Good point, thanks. The finite-volume scheme proposed in subsection 2.1 unconditionally satisfies the decay of the discrete free energy, so there is no need to impose a CFL condition linking the time step with the mesh size. This is due to the implicit nature of the finite-volume scheme, as we detail in [6].

This does not mean that the simulation works with any time step. The implicit scheme requires solving a nonlinear system of equations in every time step. Consequently, in order to find the correct convergence, the time step has to be reasonably small. In our simulations we selected $\Delta t = 0.1$ and it gave us correct convergence results with $\Delta x = \Delta y = 1$.

We have clarified this point in the manuscript. A remark about the choice of Δt , Δx and Δy has been added at the end of subsection 2.1:

Remark 2.1 (Choice of time step Δt and mesh size $\Delta x = \Delta y$). Explicit finite-volume schemes are often stable under a CFL condition imposed over the time step and mesh size. The implicit scheme presented in subsection 2.1. does not need of a CFL condition to

(a) Original image with damage

(b) $\epsilon_1 = 1.5$, $\epsilon_2 = 0.5$ and $\lambda = 1000$

(c) $\epsilon_1 = 0.5$, $\epsilon_2 = 0.5$ and $\lambda = 1000$

(d) $\epsilon_1 = 1.5$, $\epsilon_2 = 1.5$ and $\lambda = 1000$

(e) $\epsilon_1 = 1.5$, $\epsilon_2 = 0.5$ and $\lambda = 0.01$

(f) $\epsilon_1 = 1.5$, $\epsilon_2 = 0.5$ and $\lambda = 0.1$

(g) $\epsilon_1 = 1.5$, $\epsilon_2 = 0.5$ and $\lambda = 10000$

Figure 7: Sensitivity analysis for different values of ϵ_1 , ϵ_2 and λ . (A): original image with damage; (B): optimal choice of parameters; (C): short value of ϵ_1 and ϵ_2 ; (D): large value of ϵ_1 and ϵ_2 ; (E): really small value of λ ; (F): small value of λ ; (G): large value of λ .

decay the discrete free energy of the original Cahn-Hilliard equation in (1), as proved in [6]. This doesn't mean however that any Δt works in practice: in each time step the nonlinear system (11) has to be solved by iteration, and with a large Δt the convergence is likely to fail. In this work we have tested that with the choice of $\Delta t = 0.1$ and $\Delta x = \Delta y = 1$ our simulations converge to the right solution for the simulations presented in section 3.

21. Does the choice of the filter matter? Can similar accuracy be obtained with a different filter?

Our image-inpainting filter is based on solving the modified CH firstly proposed in [11]. There are more ways of implementing image-inpainting filters, and in the introduction we detail some of them: texture inpainting (exemplar-based methods), non-texture inpainting (PDEs or variational methods) or generative image inpainting (deep learning). Each method has advantages and drawbacks, and here we focus on the advantages of image-inpainting

methods based on solving PDEs such as the modified CH equation. One of the main advantage of them is that there exists fast numerical methods, such as the one we developed in [6] and apply in this work. Other filters, such as the deep-learning ones, have the disadvantage of requiring a prior dataset of images, with the benefit of being able to provide better and more general image reconstructions.

Different filters may potentially result in slight variations of accuracy. However, as long as the filter correctly removes the damage from the image, there should be a clear increase in accuracy compared to introducing the damaged image directly into the classifier.

Following your remark we have added this classification in p. 3:

“The application of other types of filters, such as texture inpainting or generative inpainting, may also result in different increases of accuracy, but overall the accuracy should be higher compared to the case where no filter is applied to the damaged image.”

22. Page 5, item 2, what is the domain of definition for x ?

The domain of definition of the ReLU activation function is all the real numbers. We have clarified this in the second item of p. 6 (before p. 5):

“2) A dense layer with 64 units and the ReLU activation function, defined as $f(x) = \max\{0, x\}$ for $x \in \mathbb{R}$.”

Appendix C

Enhancement of damaged-image prediction through Cahn-Hilliard Image Inpainting

REF. # RSOS-201294

José A. Carrillo, Serafim Kalliadasis, Fuyue Liang and Sergio P. Perez,
RESPONSE TO COMMENTS BY REVIEWER 1

We thank the Reviewer for carefully reading our manuscript and for bringing up a number of insightful comments which we feel have helped us improve its quality. We were also encouraged by the Reviewer's comment: "*I believe that present contribution deserves some merit in approaching the digital inpainting subproblem with a mathematical methodology consistent with the principles of inpainting, as opposed to leaving everything to a rather opaque neural-network framework. Overall, the manuscript is highly readable and easy to understand. The results are interesting and the paper is a good fit for the journal. I recommend publication provided the criticism and question below are addressed*".

We proceed by addressing in turn each of the Reviewer's comments, reproduced below in italics. For the sake of clarity, changes-modifications in the revised manuscript in the response to the Reviewer are highlighted in blue (while changes-modifications in response to the Associate Editor are highlighted in red).

1. Emphasis on the advantages of the finite-volume scheme is somewhat lacking. While the authors mentioned properties such as discrete energy decay, unconditionally stable, and reduction in computational costs - proven probably (since I was not able to access) in [6] for the original Cahn-Hilliard equation, it is unknown just how much can be transferred to this modified Cahn-Hilliard equation (at least not evident from the corresponding paragraphs). In particular, the modified Cahn-Hilliard equation does not admit a gradient flow structure with the Ginzburg-Landau energy.

Good point, thanks. We acknowledge that the scheme in our work in preparation [6] is for the original Cahn-Hilliard (CH) equation, and it is true that some properties such as the gradient-flow structure are not transferable to the modified CH equation. We have now remarked this on p. 3 of the Introduction:

"Furthermore, the new modified CH equation (6) is not strictly a gradient flow: although the original CH equation satisfies a gradient-flow structure under an H^{-1} norm and the fidelity term in (6) can be derived from a gradient flow under an L^2 norm, the combined modified CH equation is neither a gradient flow in H^{-1} nor L^2 ."

We have also highlighted the same point at the start of subsection 2.1:

The scheme can be straightforwardly extended to the modified CH equation in (6) proposed in [11] as we show here. Even though the modified CH equation does not possess some of the properties of the original CH equation, such as the gradient-flow structure, our finite-volume scheme preserves its robustness for all the image-inpainting test cases presented in this work. In remark 2.1 we explain how to choose the time step and the mesh size, and in remark 2.2 we detail how to turn the scheme into a dimensional splitting one, with promising applicability in high-dimensional images such as medical ones. We refer the reader to [5,6] for further details about dimensional-splitting schemes.

However, we do believe that there are crucial benefits when employing our finite-volume scheme in the modified CH equation. First, finite volumes are a straightforward discretization

when dealing with images, which often consist of rectangular-shaped pixel cells with an average color intensity. Second, the computational cost of our finite-volume scheme can be drastically reduced by employing a dimensional-splitting technique and parallelizing. This is now emphasised at the end of subsection 2.1 clarifying how the dimensional splitting provides an advantage in computational cost. Third, our scheme has been extensively tested for the original CH equation in our parallel effort work preparation [6], and even though the modified CH equation (6) is not strictly a gradient flow due to the inclusion of the fidelity term, our scheme preserves its robustness for all the image-inpainting test cases presented here. We have also added an Appendix providing a sensitivity analysis justifying the choice of the parameters λ and ϵ in order to yield an efficient finite-volume scheme for image inpainting.

We have modified the Introduction to better explain these benefits:

“In a recent effort [6] we constructed a robust semi-implicit finite-volume scheme for the CH equation that offers crucial advantages when applied to the field of image inpainting:

- Firstly, finite volumes are a straightforward discretization when dealing with images, which often consist of rectangular-shaped pixel cells with an average color intensity. This is exactly the starting point of finite-volume schemes, and as a result it is conceptually simpler to apply finite volumes in comparison with finite elements, finite differences or discontinuous Galerkin (which would be more suitable for other more complex and rare pixel shapes such as triangular ones).
- Secondly, our scheme is based on a dimensional-splitting approach: instead of solving the full two-dimensional (2D) image altogether, this technique initially solves row by row and then column by column. This has a massive benefit in computational cost, which is reduced from $\mathcal{O}(N^{d\gamma})$ for an image with N cells in d dimensions to $\mathcal{O}(dN^{d+\gamma-1})$, with $2 < \gamma < 3$. The reason for this is that the cost of inverting a $N \times N$ matrix is $\mathcal{O}(N^{d\gamma})$, with a value of $2 < \gamma < 3$ that slightly varies depending on the inversion algorithm and matrix structure (see [25] for details). For images with N cells per dimension, the solution of the full 2D scheme involves inverting a $N^d \times N^d$ Jacobian matrix, with a subsequent cost of $\mathcal{O}(N^{d\gamma})$. In contrast, the dimensional-splitting technique requires inverting dN^{d-1} Jacobians of size $N \times N$, amounting for a total computational cost of $\mathcal{O}(dN^{d+\gamma-1})$. This is already advantageous for a 2D image, and the computational cost is further reduced for high-dimensional images, such as the ones for MRI [14] or X-ray computed tomography [33] in medical image analysis. To add more, such dimensional-splitting technique allows for parallelization, and it is possible to half the total computational cost by solving nonadjacent rows and columns in parallel.
- Thirdly, our scheme has been extensively tested in [6] for challenging configurations of the original CH equation (1). In addition, in [6] we prove that the scheme unconditionally satisfies the discrete decay of the free energy for different choices of potentials (4) [13], while at the same time we prove the phase-field boundedness for mobilities of the form $M(\phi) = 1 - \phi^2$. Even though the modified CH equation (6) is not strictly a gradient flow due to the inclusion of the fidelity term, our scheme preserves its robustness for all the image-inpainting test cases presented in this work.

The combination of these properties and reduced computational cost, together with the versatility of finite volumes, make our scheme efficient and robust for the solution of the modified CH equation in (6) for a variety of applications in image inpainting.”

2. Moreover, it is not entirely convincing why practitioners might consider to use this finite-volume scheme as opposed to well-preconditioned finite-element/finite-difference methods or FFT-based spectral methods. Perhaps the advantages are less evident here due to the structure of the MNIST data set of 28 by 28 pixels, and as the authors have highlighted, the real advantages might materialise in the higher-dimensional setting.

We can see the confusion here, thank you for pointing this out. It is true that the modified CH equation is a PDE that can be solved through various numerical methods. We believe however that our finite-volume scheme has an edge over other types of schemes mainly because of two reasons: the first one is the computational cost, which can be drastically decreased thanks to a dimensional-splitting formulation coupled with parallelization. This can only be accomplished with finite-volume schemes, and for image inpainting in particular it is especially relevant in several applications such as high-dimensional medical images like MRI or X-ray computed tomography. We now extensively explain this in the Introduction (recall also our response to comment 1) as well as in a remark at the end of subsection 2.1:

“Remark 2.2 (Dimensional-splitting scheme and parallelisation). The full 2D finite-volume scheme presented in subsection 2.1 can be reformulated by employing a dimensional-splitting methodology: instead of solving the full 2D image altogether, this technique initially solves row by row and then column by column. The detailed construction of such scheme is presented in [6] and is based on [5], and here we briefly illustrate how it works.

For the dimensional-splitting approach we firstly update the solution along the x axis, for each index j corresponding to a fix value of y_j where $j \in [1, N]$. Subsequently, we proceed in the same way along the y axes, for each index i corresponding to a fixed value a value of x_i where $i \in [1, N]$. The index r , where $r \in [1, N]$, denotes the index j of the fixed y_j value in every x axis of the first loop, and the updated average density for each x axis with $j = r$ is $\phi_{i,j}^{n,r}$. Similarly, the index $c \in [1, N]$ denotes the index i for every fixed value of x_j in each y axis of the second loop, and the updated density for each y axis with $i = c$ is $\phi_{i,j}^{n,c}$.

In the first place we go through each of the x axes of the domain at a fixed y_j with $j = r$. The initial conditions for the scheme are $\phi_{i,j}^{n,0} := \phi_{i,j}^n$. The scheme for each x along the loop $r = 1, \dots, N$ satisfies

$$\phi_{i,j}^{n,r} = \begin{cases} \phi_{i,j}^{n,r-1} - \frac{\Delta t}{\Delta x} \left(F_{i+1/2,j}^{n,r} - F_{i-1/2,j}^{n,r} \right) & \text{if } j = r; \\ \phi_{i,j}^{n,r-1} & \text{otherwise;} \end{cases} \quad (1)$$

with $F_{i+1/2,j}^{n,r}$ computed in a similar fashion as in (13). Once the loop for the rows is completed, we define the intermediate density values as $\rho^{n+1/2} := \rho^{n,N}$. Subsequently, we continue through each of the y axes with index $c = 1, \dots, N$, each of them at a fixed x_i with $i = c$. The initial condition for this scheme is $\phi_{i,j}^{n,0} := \phi_{i,j}^{n+1/2}$. The scheme for each y along the loop $c = 1, \dots, N$ satisfies:

$$\phi_{i,j}^{n,c} = \begin{cases} \phi_{i,j}^{n,c-1} - \frac{\Delta t}{\Delta y} \left(G_{i,j+1/2}^{n,c} - G_{i,j-1/2}^{n,c} \right) & \text{if } i = c; \\ \phi_{i,j}^{n,c-1} & \text{otherwise.} \end{cases} \quad (2)$$

Once the loop for the columns is completed, we define the final density values $\phi_{i,j}^{n+1}$ after a discrete timestep Δt as $\phi_{i,j}^{n+1} := \phi_{i,j}^{n,N}$. This dimensional-splitting scheme can be fully

parallelized in order to save computational time. This is because: (i) the scheme does not take notice the order of updating the rows/columns, as long as all of them are updated; (ii) one row or column only depends on the values of the directly adjacent rows or columns respectively. As a result, a strategy to parallelize the dimensional splitting scheme consists of updating at the same time all the odd rows/columns, since they do not depend on one another. One can then proceed with all the even rows/columns at the same time.”

Finite volumes are a straightforward discretization when dealing with images, which often consist of rectangular-shaped pixel cells with an average color intensity. This is exactly the starting point of finite-volume schemes, and as a result it is conceptually simpler to apply them in comparison to other schemes such as finite elements, finite differences or discontinuous Galerkin (which would be more suitable for other more complex and rare pixel shapes such as triangular). This is now also added to the Introduction of the revised manuscript (again recall our response to comment 1).

3. *In the paragraph above Figure 6, the authors mentioned there are examples where the label is correctly predicted for the damaged image but incorrectly predicted for the inpainted image. Did the authors identify what went wrong for these examples? I noticed several artefacts or regions of grayscale away from the digit region appearing in the inpainted results in Figure 5. Perhaps these artefacts might be responsible for some misclassifications of the inpainted images? And if so, would some type of ad-hoc thresholding for the inpainted image, leading to a true binary image as inputs for the classification step, "fix" some of these misclassified examples?*

Good suggestion, thank you. Cases where the damaged image is correctly predicted but the inpainted image is not, occur only with high levels of damage. For instance, in Table 4 we present the accuracies depending on the number of damaged rows, and it turns out that if we damage 24 or 26 rows (out of 28 rows for MNIST) the accuracy without a Cahn-Hilliard filter is higher than with the filter (meaning that are more images correctly classified without filter than with filter). Such accuracies are already pretty low, around 20%, and we believe that the classifier is just predicting randomly.

We can very well see the confusion here, and we have accordingly modified the sentence in the paragraph above Fig. 6 to read:

“and for some of them the opposite effect can occur: that the label is correctly predicted for the damaged image but following the inpainting process it is predicted incorrectly. This only occurs for really severe damages, where there is no difference between adding the CH filter or not and the accuracies are close to the ones of a dummy classifier guessing randomly.”

We find it really interesting the proposal of adding an ad-hoc thresholding to obtain purely binary input images for the classification step. This is certainly a plausible method to investigate in future works. It is not clear though that this may help to correctly predict the misclassified examples, due to the large amount of noise contained in them. The ad-hoc thresholding could also be automatically applied by letting the image-inpainting algorithm evolve for longer times, since the only two stable wells of the modified CH equation are in -1 and 1 .

Following the comment we have added a paragraph above Fig. 6:

“In Figure 6 we also appreciate that some grayscale pixels remain in the inpainted image after applying the modified CH filter. This is due to the fact that the finite-volume scheme

has not run for long enough to turn those intensities into the stable white ($\phi = -1$) or black ($\phi = 1$) phase field. This is prone to happening for images with high levels of damage, and one should try to let the simulation run for as long as possible to avoid possible misclassification issues due to those greyscale regions.”